# CoMo: Compositional Motion Customization for Text-to-Video Generation

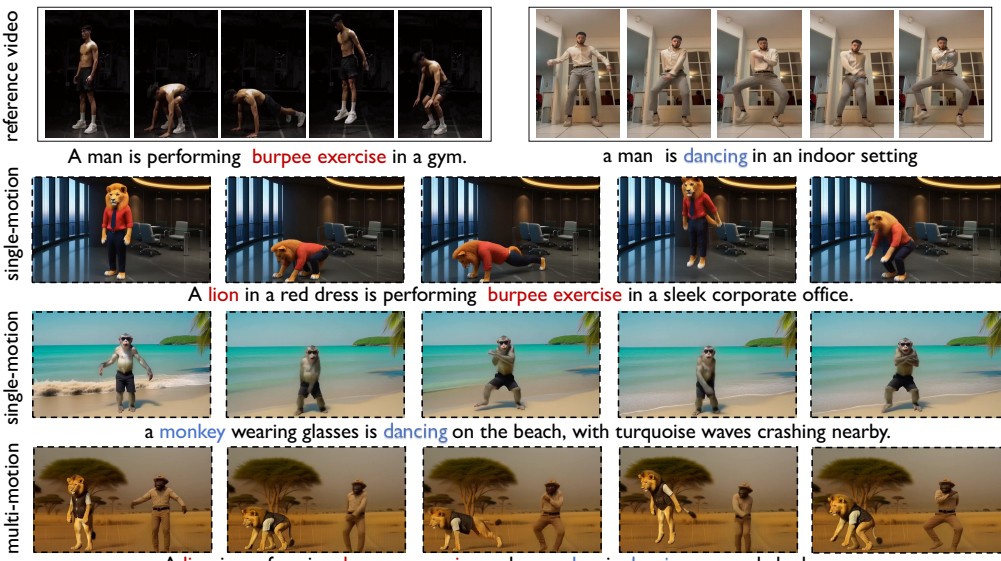

Figure 1: Our method CoMo enables learning and composing motions for text-to-video generation. The results demonstrate CoMo's effectiveness in: (a) single-motion customization, where a learned motion is transferred to a new subject and scene (*e.g.*, a lion performing a burpee in an office); and (b) multi-motion composition, where two distinct, learned motions are performed by different subjects simultaneously within the same scene.

## ABSTRACT

While recent text-to-video models excel at generating diverse scenes, they struggle with precise motion control, particularly for complex, multi-subject motions. Although methods for single-motion customization have been developed to address this gap, they fail in compositional scenarios due to two primary challenges: *motion-appearance entanglement* and ineffective *multi-motion blending*. This paper introduces CoMo, a novel framework for **compositional motion customization** in text-to-video generation, enabling the synthesis of multiple, distinct motions within a single video. CoMo addresses these issues through a two-phase approach. First, in the single-motion learning phase, a static-dynamic decoupled tuning paradigm disentangles motion from appearance to learn a motion-specific module. Second, in the multi-motion composition phase, a plug-and-play divide-and-merge strategy composes these learned motions without additional training by spatially isolating their influence during the denoising process. To facilitate research in this new domain, we also introduce a new benchmark and a novel evaluation metric designed to assess multi-motion fidelity and blending. Extensive experiments demonstrate that CoMo achieves state-of-the-art performance, significantly advancing the capabilities of controllable video generation. Our project page is at https://como6.github.io/.

## 1 INTRODUCTION

Recent Text-to-Video (T2V) models (Yang et al., 2024; Wan et al., 2025) have made tremendous progress, driven largely by an architectural shift from U-Net (Ronneberger et al., 2015) to Diffusion Transformer (DiT) (Peebles & Xie, 2023). Leveraging DiTs' superior scalability in model capacity and computational efficiency, as well as the collection of large-scale video data (Blattmann et al., 2023), state-of-the-art T2V models (Yang et al., 2024; Wan et al., 2025; Brooks et al., 2024) can synthesize diverse subjects and scenes from textual prompts. However, the ambiguity of natural language makes it difficult to convey precise motion control, often resulting in inconsistent or degraded motion generation. For example, as illustrated in Figure 1, it is impossible to animate "a lion performing a burpee exercise" with only textual prompts, which exactly mirrors the human movement in the reference video.

This motivates the task of motion customization (Zhao et al., 2024; Shi et al., 2025; Yatim et al., 2024). Specifically, they aim to adapt pre-trained T2V models to generate new videos that replicate the motion from a reference video, while maintaining the flexibility to create diverse subjects and scenes. Prior methods (Zhao et al., 2024; Ren et al., 2024) typically first learn or extract the motion pattern from the reference video, and then transfer it to a new subject in the generated video. Although these methods have demonstrated promising results, they mainly focus on *single-motion* customization. Thus, they can only apply one single learned motion pattern into generated videos (*e.g.*, a monkey is dancing or a lion is performing burpee). A more complex and practical challenge remains unexplored: ***Compositional Motion Customization***. It requires composing multiple, distinct motions within the generated video, such as composing dancing and burpee into the same video (*c.f.*, Figure 1). Furthermore, pretrained T2V foundation models often struggle to reliably generate complex scenes involving subjects with different motions. Therefore, tackling this novel setting enables the generation of richer and more dynamic content that better aligns with the complexity of real-world scenarios.

Despite this growing demand, naively extending existing single-motion customization methods for this new setting faces two major challenges:

- **Motion-Appearance Entanglement.** In the reference video, the relationship between motion and appearance is intricately entangled. During the learning of motion patterns, the model may inadvertently memorize irrelevant appearance characteristics, thereby compromising its ability to generalize learned motion to subjects with different visual appearances. This challenge is substantially amplified in the compositional setting. When learning multiple motions from different reference videos, the model must disentangle several motion-appearance pairs simultaneously. To mitigate this issue, previous methods (Zhao et al., 2024; Ren et al., 2024) adopt a decoupling strategy within 3D U-Nets by isolating and optimizing motion-specific temporal attention modules. However, this reliance on separable attention is fundamentally incompatible with modern DiT-based models, whose unified attention structure poses a significant adaptation challenge. Moreover, these approaches are designed for single-motion contexts and are not equipped to handle the compositional challenges described next.
- **Multi-Motion Blending.** A primary challenge in compositional motion is motion blending, where distinct motions assigned to different subjects corrupt each other. Current motion customization methods (Zhao et al., 2024; Ren et al., 2024) lack a sophisticated mechanism to bind specific motions to their intended subjects or spatial regions. To compose the multiple motions, they resort to naive strategies such as: (i) Linearly merging the parameters of separately learned motion modules; or (ii) Jointly training a single model on a collection of reference videos, where each video demonstrates one of the desired motions. Both approaches typically result in an incomplete and distorted fusion of movements (*c.f.*, Figure 2).

In this paper, to address these challenges, we propose ***CoMo***, the first framework capable of precisely composing multiple motions into one generated video. Built upon the recent state-of-the-art DiT model (Wan et al., 2025), CoMo adopts a two-phase design that directly addresses the two challenges respectively: 1) decoupled single-motion learning and 2) plug-and-play multi-motion composition. In the first phase, we learn each motion pattern separately, resulting in a dedicated single-motion module for every motion. In the second phase, these modules are seamlessly integrated to generate multi-motion videos without requiring additional training.

**In the single-motion learning phase**, we employ a static-dynamic decoupled tuning paradigm to disentangle the motion and appearance. We utilize two separate Low-Rank Adaptation (LoRA) (Hu

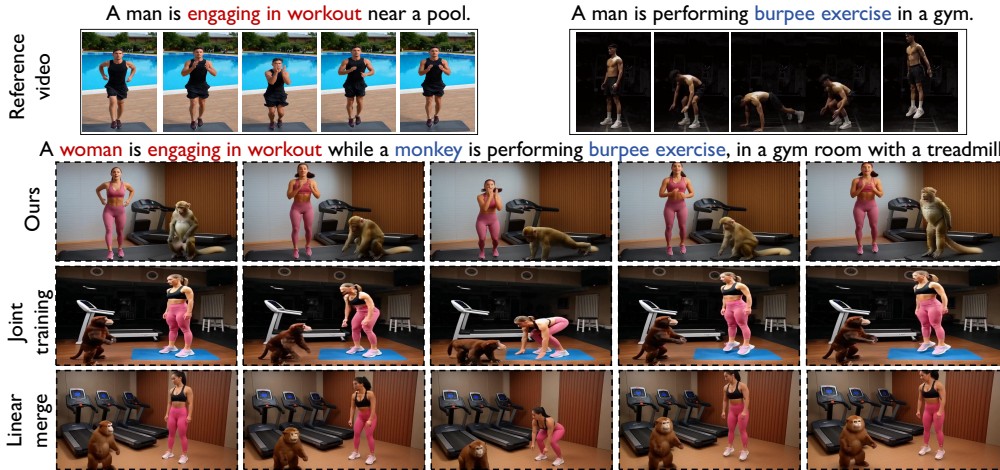

Figure 2: Comparison of motion blending capabilities. We evaluate CoMo's ability to compose two distinct motions from different reference videos into a single scene with new subjects. Compared to baseline methods (*e.g.*, linear merging and joint training) have incomplete and distorted movements, CoMO successfully generates a coherent video where both actions are performed naturally and simultaneously, preserving the integrity of each motion.

et al., 2022) modules for motion and appearance, respectively. We first train the static LoRA module on the unordered video frames to absorb the appearance characteristics of the reference video. Subsequently, the static LoRA is frozen, and a separate dynamic LoRA module is optimized to reconstruct whole video clip. This sequential tuning strategy enforces the dynamic LoRA focus solely on capturing the motion pattern without absorbing the appearance features. Due to this model-agnoistic design, our learning method can be utilized in different architectures (*e.g.*, DiT). **In the multi-motion composition phase**, we propose a simple yet effective strategy, termed *divide-and-merge*, to guide the video generation process. Specifically, we first divide the global latent into distinct sub-regions, each of which is denoised separately by the motion-specific dynamic LoRA from phase one. Subsequently, the predicted subregional noise is merged through Gaussian weight for updating the generation directions. By spatially isolating the influence of each motion-specific module, our method prevents the parameter-level interference caused by linear merging and avoids the motion-blending ambiguity that arises from joint training. As evidenced in Figure 1, our method can generalize the single motion to diverse subjects across various scenes. And with the divide-and-merge strategy, it successfully generates videos exhibiting multiple motions.

As a pioneering effort in this direction, we further collected a new benchmark for both single and compositional motion customization. Moreover, given the absence of efficient evaluation metrics for motion composition in existing methods, we introduce a novel metric inspired by MuDI (Jang et al., 2024) to assess the multi-motion fidelity while taking into account the degree of motion blending. Extensive experiments have demonstrated that CoMo achieves state-of-the-art performance in both single and composition motion customization. In summary, our contributions are as follows:

- We are the first to identify and tackle the challenging task: compositional motion customization, which aims to synthesize multiple distinct motions within a single video, pushing the boundaries of controllable video generation.
- We propose CoMo, a novel and effective two-phase framework for this new task. It features a static-dynamic decoupled tuning paradigm to learn pure motion patterns and a plug-and-play divide-and-merge strategy for composing multiple motion patterns.
- To facilitate systematic evaluation in this new area, we introduce a new benchmark and evaluation metric. The designed metric can not only assess the fidelity of multi-motion synthesis, but also account for motion blending.

## 2 RELATED WORK

**Video Motion Customization.** Motion Customization aims to replicate the motion from a reference video within a generated video. The primary challenge of this task is the entanglement be-

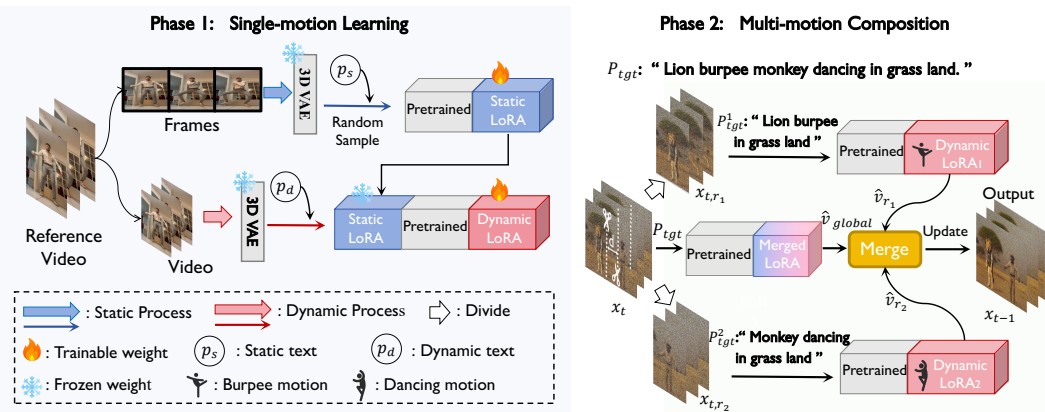

Figure 3: **An overview of our proposed CoMo framework.** CoMo consists of two phases: (1) decoupled single-motion learning and (2) plug-and-play multi-motion composition. In Phase 1, we first train a static LoRA module on random frames to learn the appearance of the reference video. Then, we freeze the static LoRA and train a dynamic LoRA module on the complete video to exclusively capture its motion patterns. In Phase 2, we introduce a divide-and-merge strategy for compositional motion generation, while all the weights are frozen.

tween motion and appearance in the source video. When learning a motion pattern, the model tends to memorize irrelevant appearance features from the reference, thereby compromising its ability to generalize the learned motion to subjects with different visual appearances. Prior methods (Zhao et al., 2024; Ren et al., 2024; Jeong et al., 2024; Materzyńska et al., 2024), primarily built on U-Net architectures, utilized factorized spatial and temporal attention modules to decouple appearance and motion. However, this reliance on separable attention is fundamentally incompatible with modern DiT-based models (Yang et al., 2024; Wan et al., 2025; Brooks et al., 2024), which employ a unified spatiotemporal attention structure. While recent works (Ma et al., 2025; Shi et al., 2025; Pondaven et al., 2025) have begun to adapt DiT-based models for single-motion customization, they do not address the more complex challenge of compositional motion customization, which requires composing multiple, distinct motions within a single generated video. To our knowledge, our work is the first to address this compositional challenge by proposing a plug-and-play composing strategy during inference time.

## 3 APPROACH

**Problem Formulation.** Given a set of reference videos $V_{ref} = \{V_{ref}^1, V_{ref}^2, ... V_{ref}^n\}$, where each video $V_{ref}^i$ showcases a distinct motion denoted as $M_i$, and a user-provided prompt $P_{tgt}$ describing a new scene with new subjects. Our goal is to generate a video based on $P_{tgt}$ in which: 1) the appearance of the subjects and the background scene are controlled by the text prompt $P_{tgt}$, and 2) each motion $M_i$ is transferred to the corresponding subject as specified in $P_{tgt}$.

**General Framework**: As shown in Figure 3, our proposed CoMo consists of two phases: **1)** Decoupled single-motion learning (Sec 3.1): for each reference video, we first train a static LoRA ($\Delta\theta_s$) to absorb the appearance characteristics, and then optimizing a dynamic LoRA ($\Delta\theta_d$) to capture the specific motion represented in the reference video. **2)** Plug-and-play multi-motion composition (Sec 3.2): Instead of a naive combination, we present a unique divide-and-merge scheme to compose the learned motion-specific dynamic LoRA for customizing multi-motion video generation.

### 3.1 SINGLE-MOTION LEARNING

Given a reference video $V_{ref}^i$ representing a specific motion $M_i$, this phase aims to learn the particular motion pattern.

**Video Preprocessing.** As a prerequisite for decoupled tuning, we first extract all frames from $V_{ref}^i$. Using a pre-trained 3D VAE encoder (Kingma & Welling, 2013), we encode the individual frames into a set of frame latents, denoted as $U(V_{ref}^i)$. The entire video clip is also encoded into a single

video latent $x_d$. Following (Abdal et al., 2025), we employ two distinct text prompts to guide the learning process: a static prompt ($p_s$) and a dynamic prompt ($p_d$), which is motion-descriptive. These prompts are automatically generated using a video captioning model (Hong et al., 2024).

**Decoupled Tuning.** Since the pre-trained video DiT model is built upon a unified attention structure, which jointly models spatial and temporal information. Directly fine-tuning the LoRA on a single video may capture motion and appearance features simultaneously. Inspired by previous work (Abdal et al., 2025), we adopt a static-dynamic decoupled tuning approach to explicitly separate the motion and appearance into distinct LoRA weight spaces. Our decoupling strategy involves a sequential two-step optimization process for each reference video.

*1) Static Appearance Learning.* To exclusively capture the appearance, we train a static LoRA ($\Delta\theta_s^i$) using frames randomly sampled from the reference video. The objective is defined as:

$$\mathcal{L}_s = E_{x_s \sim U(V_{ref}^i), x_0 \sim N(0,I)} \left\| v_t - v_{(\theta + \Delta\theta_s^i)}(x_t, t, p_s) \right\|_2^2, \tag{1}$$

where $x_s$ is the randomly sampled frame latent of the latents set, $x_t$ is the noisy latent of $x_s$, $\theta$ is the frozen base model parameters, and $p_s$ is the **description of the appearance** (*e.g.*, "a photo of a man in the indoor setting").

*2) Dynamic Motion Learning.* Once the static LoRA $\Delta\theta_s^i$ gets converged, we introduce an independent dynamic LoRA ($\Delta\theta_d^i$) to capture the motion. During this phase, we freeze the parameters of both the base model $\theta$ and trained static LoRA $\Delta\theta_s^i$. The optimization is performed on the full video sequence $V_{ref}^i$, with only the parameters of $\Delta\theta_d^i$ being updated. The objective function is:

$$\mathcal{L}_d = E_{x_d, x_0 \sim N(0,I)} \left\| v_t - v_{(\theta + \Delta\theta_s^i + \Delta\theta_d^i)}(x_t, t, p_d) \right\|_2^2. \tag{2}$$

Here, $x_t$ is the noisy latent of the video latent $x_d$ instead of the frame latent, and $p_d$ is the corresponding motion-descriptive prompt (*e.g.*, "a man is performing burpees"). Since the static LoRA provides the necessary appearance information, the optimization process forces the network to encode the residual information — the temporal dynamic into the weights of $\Delta\theta_d^i$. This procedure effectively isolates the motion $M_i$ into a compact, dedicated parameter space, making it an independent and composable motion representation.

**Context-Agnostic Motion Prompting.** Motion concepts often exhibit intrinsic entanglement with contextual priors (*e.g.*, "snowboard" implicitly associates with snowy terrains). Using "snowboarding" learn such motion risks absorbing scene-specific biases, causing semantic conflicts when transposed to novel scenes(*e.g.*, "snowboarding on desert" yields implausible visuals). To mitigate this issue, we reformulate the motion-descriptive prompt $p_d$ to focus on the fundamental kinematics of the action (*e.g.*, "sliding on a snowboard")[2]. This granular decomposition decouples the motion from its typical environment, creating a more generalizable representation.

## 3.2 MULTI-MOTION COMPOSITION

Given multiple motion-specific LoRAs $\{\Delta\theta_d^i\}_{i=1}^N$, where each $\Delta\theta_d^i$ is the dynamic LoRA trained after the first phase[1]. This phase aims to generate a multi-motion video, where every motion is represented by $\Delta\theta_d^i$. To achieve this, we propose a plug-and-play divide-and-merge (DAM) strategy. This strategy effectively composes multiple motion-specific LoRAs by incorporating Gaussian Smooth Transition for local blending and Global Blending for overall consistency.

**Divide-and-Merge.** Inspired by image composition methods (Bar-Tal et al., 2023; Jiménez, 2023), DAM operates at each step of the denoising process. The core idea is to divide the global latent space into distinct sub-regions, guide each with a specific motion LoRA, and then merge the resulting velocity predictions to form a coherent global update. However, a naive merge can create visible seams and inconsistent video content. To address this issue, DAM incorporates two key techniques: Gaussian Smooth Transition for local seamlessness and Global Blending for overall consistency.

*1) Divide.* As shown in Figure 3, at each denoising step $t$, we partition the global video latent $x_t$ into $N$ rectangular regions $\{r_1, ..., r_N\}$. For each region $r_i$, we use its corresponding motion-specific

---

[1]Due to method's flexibility, motion-specific LoRAs can come from external repositories, *e.g.*, civitai.com.

LoRA module $\Delta\theta_d^i$ and a targeted text prompt $P_{tgt}^i$[2] to predict a local velocity field. This isolates the guidance for each motion to its intended spatial area. The velocity prediction for a single region $x_{t,r_i}$ is calculated using classifier-free guidance (Ho, 2022):

$$\hat{v}_{r_i} = (1 + s_i)\, v_{(\theta+\Delta\theta_d^i)}\left(x_{t,r_i}, t, P_{tgt}^i\right) - v_{(\theta+\Delta\theta_d^i)}\left(x_{t,r_i}, t, \emptyset\right), \tag{3}$$

where $s_i$ is the guidance scale for region $r_i$, and $\theta$ is the frozen weight of the base model.

*2) Merge.* The individual regional predictions $\hat{v}_{r_i}$ must be merged back into a single global velocity field to update the global latent $x_t$. We achieve this through a two-stage process that ensures both local and global coherence (*c.f.*, Figure 4).

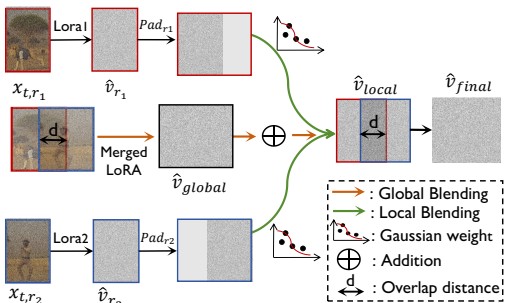

**Gaussian Smooth Transition for Local Blending.** To prevent sharp boundaries between adjacent regions, we first ensure the regions have a slight spatial overlap (d latent units in Figure 4). We then merge the local velocity predictions using a weighted mixture, where the weights ensure a smooth fall-off near the boundaries. Specifically, the weight matrix $w_i$ for each region $r_i$ is generated from a bivariate Gaussian distribution centered within that region. This makes the influence of each regional model stronger at its center and gracefully weaker towards its edges. The merged velocity field $\hat{v}_{local}$ is formed as:

Figure 4: **The details of the divide-and-merge strategy.** Final velocity prediction is generated by merging regional velocities with a Gaussian smooth transition for local blending, followed by a global blending step to ensure overall consistency. Here, we omit the time dimension for simplicity.

$$\hat{v}_{local} = \frac{1}{\sum_i w_i} \odot \sum_{i=1}^{N} w_i \odot Pad_{r_i}\left(\hat{v}_{r_i}; x_t\right). \tag{4}$$

$Pad_{r_i}(\cdot)$ is a padding operation that places the regional prediction $\hat{v}_{r_i}$ into a global tensor of the same size as $x_t$ and fills the outside with zeros. Element-wise product $\odot$ applies Gaussian weights, and the denominator normalizes the result, ensuring a seamless blend in the overlapping zones.

**Global Consistency Blending.** While Gaussian smoothing handles local transitions, it may not guarantee overall scene consistency. To address this, we introduce a global blending step. We compute a globally coherent velocity prediction, $\hat{v}_{global}$, by using a single model where all motion-specific LoRAs are linearly averaged ($\overline{\Delta\theta_d} = \frac{1}{N}\sum_i \Delta\theta_d^i$) and guided by the global prompt $P_{tgt}$. Final velocity field $\hat{v}_{final}$ is a linear interpolation between global prediction and locally merged one:

$$\hat{v}_{final} = \lambda \cdot \hat{v}_{global} + (1 - \lambda) \cdot \hat{v}_{local}. \tag{5}$$

Here, $\lambda$ is a hyperparameter that controls the strength of the global consistency. Finally, we utilize the blending velocity predictions $\hat{v}_{final}$ to update the noisy video latent $x_t$. In practice, we find that it is most effective to apply this global blending during the initial $\tau$ denoising steps. The complete procedure for our multi-motion composition is detailed in Algorithm 1.

## 3.3 NEW METRIC FOR MULTI-MOTION FIDELITY

Existing motion fidelity metrics (Yatim et al., 2024; Shi et al., 2025) are designed for single-motion tasks and are ill-suited for compositional generation, as they overlook the multi-motion blending. To address this, we introduce the *Crop-and-Compare* (C&C) framework, inspired by MuDI (Jang et al., 2024), for robust multi-motion evaluation. As illustrated in Figure 5, C&C first isolates individual motions from the generated video into cropped videos $\{B_i\}_{i=1}^{N}$ using OWLv2 (Minderer et al., 2023) and SAM2 (Ravi et al., 2024), where each video $B_i$ corresponds to a reference video $V_{ref}^i$. We then compute two matrices: a ground-truth similarity matrix $S^{GT}$ between all pairs of reference videos ($V_{ref}^i$ and $V_{ref}^j$), and a C&C similarity matrix $S^{CC}$ between each cropped clip $B_i$ and every reference video $V_{ref}^i$. The final C&C score is defined as $1 - ||S^{CC} - S^{GT}||$. A higher score indicates that $S^{CC}$ closely matches the ideal similarities in $S^{GT}$, signifying high fidelity for each

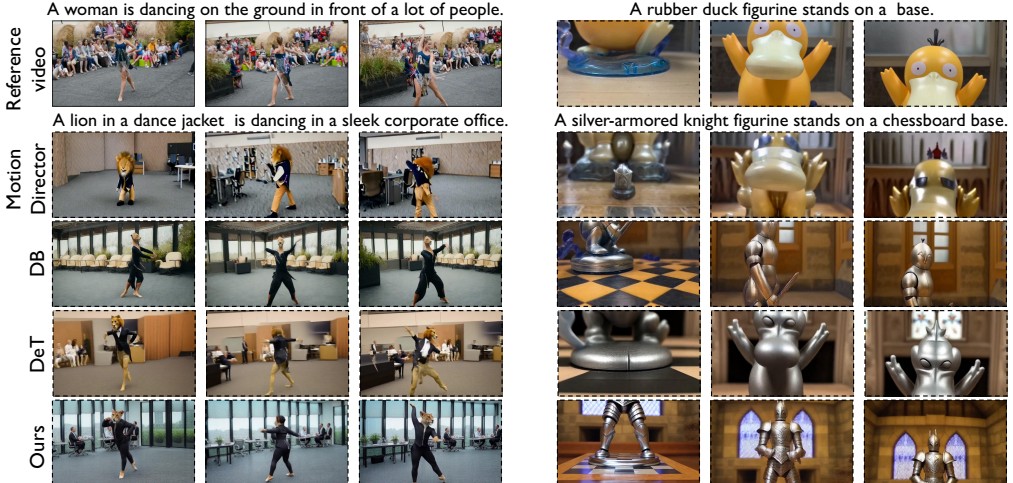

Figure 5: **Overview of the Crop-and-Compare**. We calculate the motion fidelity between the cropped video and the reference video and compare $S^{GT}$ and $S^{CC}$ to evaluate multi-motion fidelity.

Figure 6: **Qualitative comparison of single motion customization.** Our method outperforms all baselines in precisely capturing the motion and maintaining edit fidelity.

distinct motion (diagonal entries) and minimal blending between them (off-diagonal entries) [2] we provide qualitative examples in Figure 17 and human evalutaions in Table 3.

## 4 EXPERIMENT

### 4.1 EXPERIMENTAL SETUP

**Evaluation Dataset.** For the single motion customization, we curated a diverse dataset comprising videos from the Internet and previous studies (Abdal et al., 2025; Shi et al., 2025; Zhao et al., 2024). This dataset is composed of a wide range of motion types, including complex human motion, object motion, and camera motion. To evaluate the effectiveness of our method for composing multi-motion, we randomly select motions for composition. The selected motions include complex human motion, object motion, etc.

**Baselines.** For the single motion customization, we compare our method with state-of-the-art motion customization methods, including DeT(Shi et al., 2025), 3D U-Net-based method MotionDirector(Zhao et al., 2024), and DreamBooth (DB)(Ruiz et al., 2023). For the compositional motion customization, we compare against several strategies: a naive joint-training DreamBooth baseline, a direct linear merge of our learned motion LoRAs, and VACE (Jiang et al., 2025), a unified video generation and editing framework. We implement DreamBooth on the dit model (Wan et al., 2025) to ensure a fair comparison.

### 4.2 QUALITATIVE RESULTS

**Single Motion Customization.** As illustrated in Figure 6, our method excels at generalizing a learned motion to novel subjects with disparate appearances, such as transferring a human's motion

---

[2]Due to the limited space, more results and details are left in the Appendix..

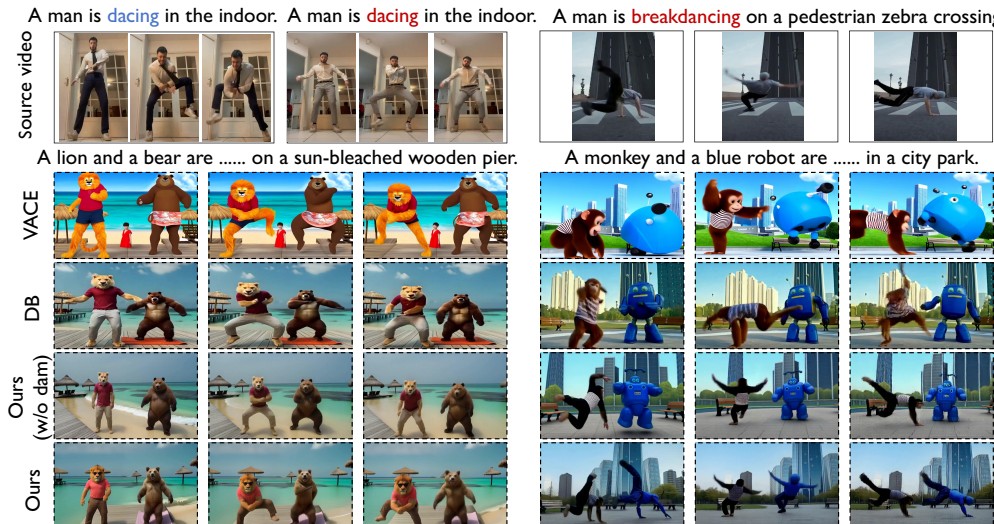

Figure 7: **Qualitative comparison of compositional motion customization.** Our method can assign different subjects to different or the same motions.

Table 1: **Quantitative and user study results with SOTA video motion customization methods.**

| Method | Quantitative Metrics | | | | User Study | | | |
|---|---|---|---|---|---|---|---|---|
| | Text Sim.↑ | Motion Fid.↑ | Temp. Cons. | C&C.↑ | App.↑ | Motion Pres.↑ | Temp. Cons.↑ | Overall↑ |
| **Single-Motion Customization** | | | | | | | | |
| MotionDirector | 0.421 | 0.785 | 0.958 | | 46 | 94 | 61 | 56 |
| DreamBooth | 0.446 | 0.762 | 0.970 | | 144 | 117 | 140 | 134 |
| DeT | 0.456 | 0.813 | **0.973** | | 109 | 138 | 130 | 131 |
| Ours | **0.470** | **0.865** | 0.967 | | **401** | **351** | **369** | **379** |
| **Compositional Motion Customization** | | | | | | | | |
| Base Model | 0.485 | | **0.981** | | | | | |
| DreamBooth | 0.469 | 0.349 | 0.980 | 0.349 | 186 | 166 | 199 | 177 |
| VACE | 0.488 | 0.494 | 0.978 | 0.367 | 343 | 322 | 336 | 343 |
| Ours (w/o dam) | 0.475 | 0.659 | 0.971 | 0.473 | 146 | 171 | 164 | 137 |
| Ours | **0.488** | **0.663** | 0.973 | **0.592** | **585** | **601** | **561** | **603** |

to a lion. In comparison, competing methods exhibit significant limitations. MotionDirector struggles with maintaining motion fidelity and demonstrates poor text editing fidelity. We observe that DreamBooth suffers from overfitting, and it is hard to transfer the learned motion to a new appearance. While DeT can capture the foreground motion, it often suffers from absorbing the appearance of the foreground, as shown in the right panel of Figure 6.

**Compositional Motion Customization.** As shown in Figure 7, our proposed divide-and-merge strategy demonstrates remarkable flexibility in assigning distinct motions to multiple subjects and the same motion to different subjects. Conversely, baseline approaches fail to achieve coherent multi-motion synthesis. Both the joint training of DreamBooth and the linear merging of LoRAs result in significant motion interference, where distinct movements blend and corrupt one another over the video's duration (e.g., the conflicting dance motions in Figure 7). VACE struggles to capture accurate motion patterns, particularly for complex or occluded actions. In stark contrast, our method precisely composes multiple motions in their designated regions and provides robust text-based control over both subject and background appearance. Furthermore, our method can be applied to more subjects and motions flexibly (*c.f.*. Figure 16 in the Appendix).

### 4.3 QUANTITATIVE RESULTS

**Numerical Comparison.** To evaluate our method, we conduct quantitative comparisons for both single-motion and compositional motion customization tasks. For the single-motion setting, we assess performance using three standard metrics. (1) Text Similarity: Following previous works (Zhao et al., 2024; Shi et al., 2025), we measure the average CLIP score (Radford et al., 2021) between the generated frames and the target text prompt. (2) Temporal Consistency: We evaluate frame-to-

frame coherence by computing the average CLIP feature similarity between consecutive frames. (3) Motion Fidelity: Adopting the protocol from (Yatim et al., 2024), we compute a holistic similarity score between motion tracklets extracted from the reference and generated videos. For the more challenging compositional motion setting, we utilize our proposed $C\&C$ score to specifically evaluate multi-motion fidelity. In addition, we compute a standard motion fidelity score by averaging the motion fidelity between the generated video and each of the multiple reference videos. As presented in Table 1, our method demonstrates superior performance across both scenarios.

**Human Evaluation.** We perform the human evaluation to reflect real preferences on the generated videos. Specifically, we invited 35 volunteers and gave them a reference video (multiple reference videos for multi-motion setting), a target prompt, and four target videos generated by different models. They are asked to choose the best target videos that they believe demonstrate the best results across four aspects–motion preservation, appearance diversity, video smoothness, and overall quality. As shown in Table 1, our method is preferred over baselines in both settings.

### 4.4 ABLATION STUDY

In this section, we conduct a systematic ablation study to isolate and quantify the contribution of each key component in our framework. The qualitative and quantitative ablation study results are shown in Figure 8 and Table 2, respectively[2].

**Necessity of Decoupled Tuning.** To validate our decoupled tuning strategy, we train a single LoRA module directly on the entire reference video, forcing it to learn both appearance and motion simultaneously. As shown in the third column of Figure 8 and Table 2, this joint tuning approach leads to severe appearance leakage and fails to isolate a clean motion pattern. Consequently, this impure motion representation is detrimental to the composition phase, resulting in significant motion blending and distorted outputs.

**Importance of Gaussian Smooth Transition.** We evaluate the role of our local blending mechanism by removing the Gaussian smooth transition and instead performing a naive

Table 2: Quantitative ablation.

| Method | Text Sim ↑ | Motion Fid ↑ | Temp. Cons. | C&C ↑ |
|---|---|---|---|---|
| w/o DT | 0.475 | 0.557 | 0.975 | 0.514 |
| Ours | 0.488 | 0.663 | 0.971 | 0.592 |

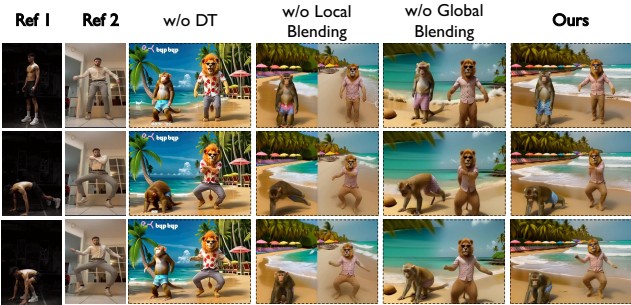

Figure 8: **Ablation study about proposed modules**. We remove the proposed modules to evaluate their effectiveness. "DT" means decoupled tuning in phase 1.

merge of the regional velocity predictions. The result depicted in the fourth column of Figure 8 and the "w/o Local Blending" exhibits jarring artifacts and visible seams at the boundaries between different motion regions. This "hard merge" disrupts the spatial coherence of the video, creating an unnatural collage effect and validating the necessity of our smooth blending technique.

**Importance of Global Consistency Blending.** Next, we assess the impact of the global consistency blending by removing it (*i.e.*, setting $\lambda = 0$). While Gaussian smoothing alone can handle local transitions, the lack of a global prior results in inconsistencies in the overall scene, such as conflicting lighting or background textures across different regions (fifth column of Figure 8).

## 5 CONCLUSIONS

In this paper, we introduce and address the novel and challenging task of compositional motion customization. Our proposed framework effectively tackles the core challenges of motion-appearance entanglement and multi-motion blending that hinder existing methods. Through a two-phase approach, CoMo first isolates pure motion patterns using a static-dynamic decoupled tuning strategy and then seamlessly composes them using a plug-and-play divide-and-merge technique. This allows for the precise synthesis of multiple, distinct motions within a single, coherent video, which is

not explored by prior work. Furthermore, we established a new benchmark and a novel evaluation metric ($C\&C$ score) to facilitate rigorous and meaningful assessment in this new research direction. Extensive experiments demonstrate that CoMo not only excels at single-motion customization but also sets a new state-of-the-art in the compositional setting. We believe this work opens up new possibilities for creating richer, more dynamic, and highly controllable video content.

**Ethics statement.** This work does not involve ethical issues and poses no ethical risks.

**Reproducibility Statement.** We make the following efforts to ensure the reproducibility of our proposed method: (1) Our training and inference codes together with the trained model weights will be publicly available. (2) We provide training and inference details in the appendix (Sec. A). (3) The reference videos of our new benchmark will be publicly available. (4) We provide more analysis and discussion in the appendix (Sec. D).

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

# Appendix

**Overview.**

## A    IMPLEMENTATION DETAILS

**Details of the benchmark.** We evaluate our framework on a curated benchmark of 23 high-quality reference videos ( from previous works like dynamic concepts(Abdal et al., 2025) and platforms like TikTok), characterized by high kinematic diversity. The dataset is primarily grounded in human-centric videos (19 clips), as these motion exhibits intricate articulation and complexity ranging from diverse street dancing and athletic sports to challenging dynamic poses like handstands. These complex sequences rigorously test the model's ability to decouple dynamics from appearance. To further validate broad generalization beyond human subjects, our dataset also explicitly includes distinct 2 object motions and 2 camera movements (e.g., circling views). For multi-motion setting, we constructed 18 compositional pairs (e.g., "`burpee and dancing`"). Note that the composed motions have the same temporal length for simplicity. For evaluation, to compare our method in single motion setting, we generated 5 videos per motion (115 videos total) for testing. In multi-motion setting, we generated 5 videos per pair (90 videos total). The full dataset, along with the prompt list, will be open-sourced.

**Training details.** In our experiment, we utilize the open-source video generation model WAN-2.1(Wan et al., 2025) as the foundational text-to-video generation model. Our method only needs to be trained in the single-motion learning phase, and we train the LoRA modules using the Adam optimizer with default beta values ($\beta_1 = 0.9$, $\beta_2 = 0.95$) and an epsilon of 1e-8. We employ a learning rate of 5e-5 and a weight decay of 1e-4. A dropout rate of 0.2 is applied during training to mitigate overfitting. The ranks for the static and dynamic LoRA modules are set to 256 and 512, and the LoRA weights are injected into the query, key, value, and output linear layers. The static LoRA is trained for 1000 iterations, while the dynamic LoRA's training duration varies based on motion complexity, ranging from 800 steps for simple movements (e.g., a riding horse) to 1500 steps for complex actions (e.g., breakdancing).

**Inference details.** During the composition phase, each spatial region $r_i$ is defined as a square with side lengths equal to the height of the video latent. The overlapping region between adjacent areas is set to a height of 32 latent units. The global blending hyperparameter $\lambda$ is set to 0.5, and this blending operation is applied only during the initial 10 denoising steps. For inference, we leverage the flow matching scheduler (Lipman et al., 2022) with a sampling step of 50, and the classifier-free guidance (CFG) scale is consistently set to 6.0 for all generations.

**Baselines implementation details.** For single-motion comparisons, we implement DreamBooth by training a single rank-512 module for 2500 steps, injecting weights into the query, key, value, and output layers. For other methods like MotionDirector, we adhere to their official implementations and default settings. For compositional motion tasks, the DreamBooth baseline is jointly trained on all reference videos for 3200 steps with a LoRA rank of 512. The VACE baseline is guided by a composite signal created by extracting and concatenating pose sequences from each reference video. Finally, as an ablation, we evaluate a variant of our method that omits our divide-and-merge strategy and instead linearly merges the learned LoRA modules from the first phase.

# B    MORE QUALITATIVE RESULTS

We show more single-motion qualitative results in Figure 9 and Figure 10. Each source video is combined with four newly generated videos. We also provide more compositional motion qualitative results in Figure 16, Figure 18 and Figure 19. Due to the flexibility of our provided ivide-and-merge strategy, we can assigning distinct motions to multiple subjects or the same motion to different subjects.

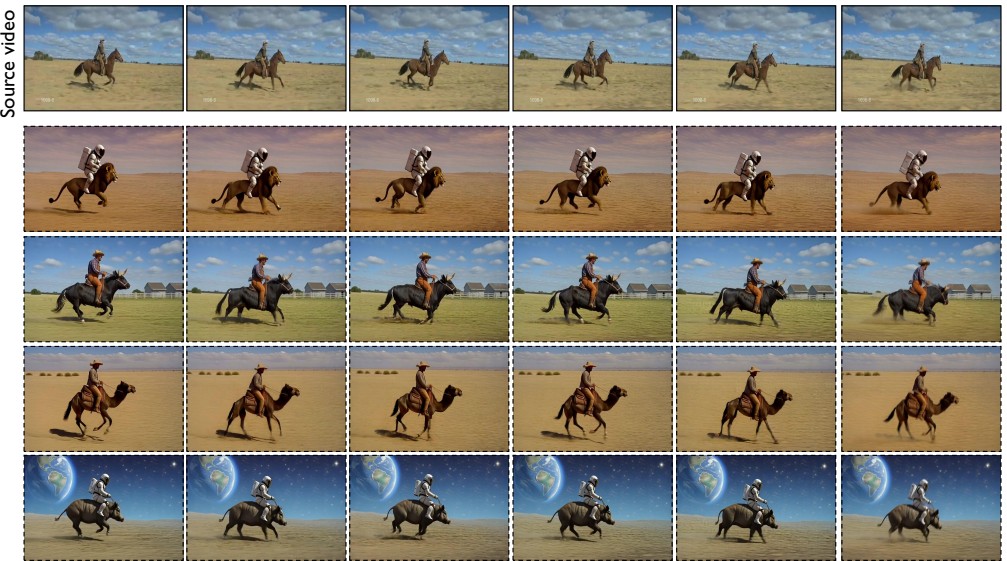

Figure 9: **More Single Motion Customization Results.** Each source video is combined with four newly generated videos.

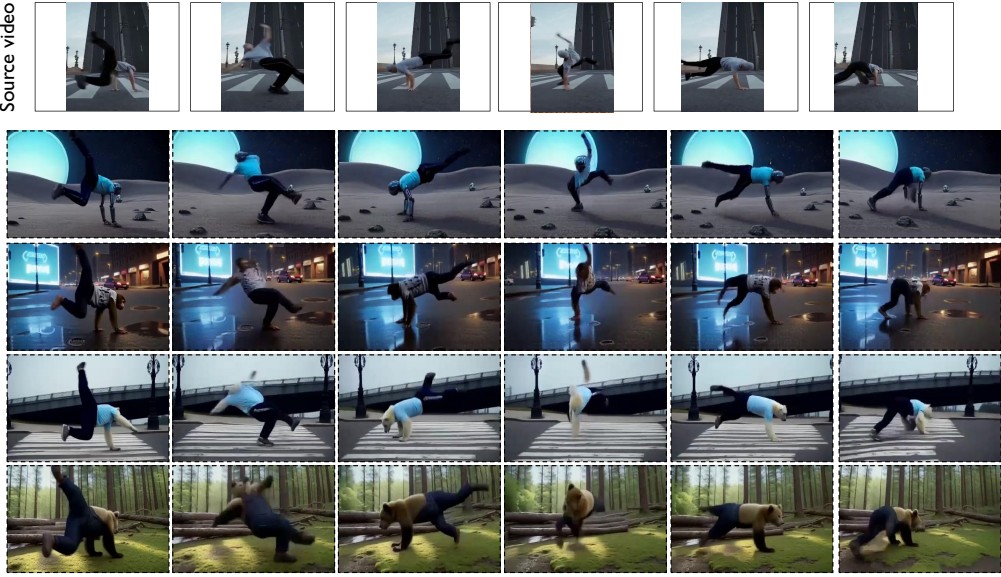

Figure 10: **More Single Motion Customization Results.** Each source video is combined with four newly generated videos.

# C    MORE QUALITATIVE COMPARISON RESULTS

We present additional single-motion qualitative comparisons in Figure 11 and Figure 12. Our method excels at generalizing a learned motion to novel subjects with disparate appearances. We also show additional compositional motion qualitative comparisons in Figure 14 and Figure 15. Our

proposed method can accurately assign distinct motions to the specific subject, showing the remarkable flexibility of our method.

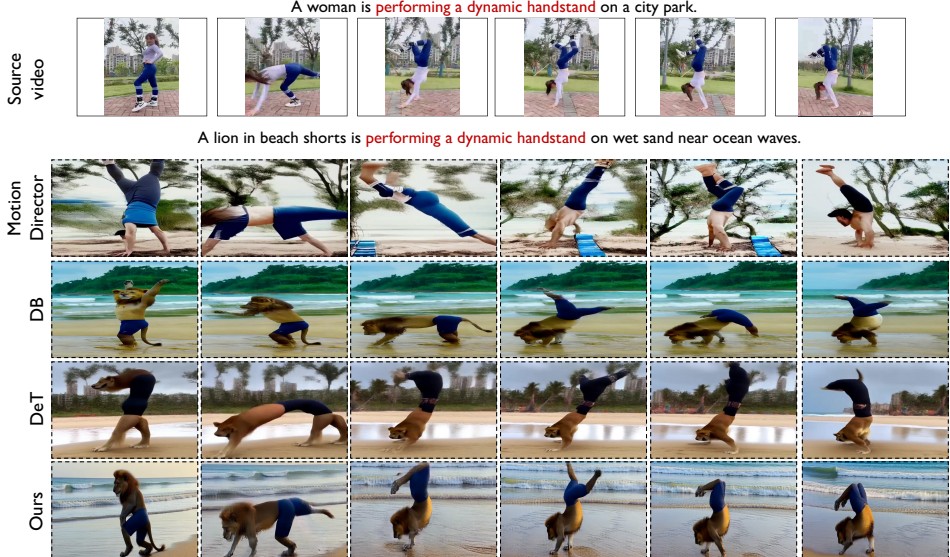

Figure 11: **More Single Motion Qualitative Comparison Results.**

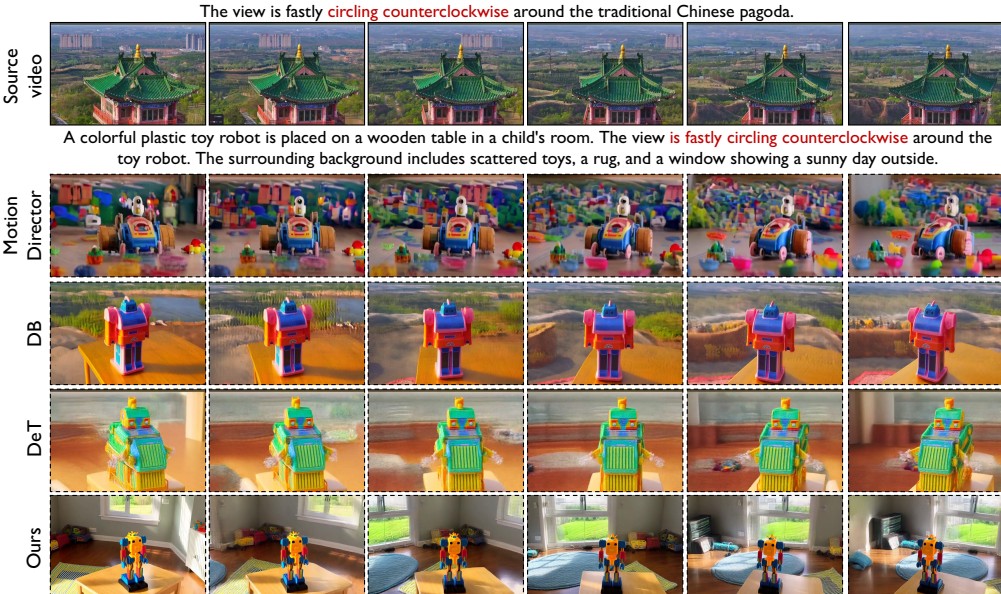

Figure 12: **More Single Motion Qualitative Comparison Results.**

## D    ADDITIONAL ANALYSIS AND ABLATION STUDY

**Composition of More Motions.** To demonstrate the scalability and flexibility of our divide-and-merge strategy, we test its performance on a more complex task that has three distinct motions involving three and four subjects, respectively (the 2rd and 3rd row in Figure 16). We also test its performance on composing the same motion in Figure 18. Our method successfully composes all three motions without significant quality degradation or motion interference (left panel of Figure 16). In stark contrast, the base model, Wan (Wan et al., 2025), fails to generate text-aligned videos when handling multiple subjects (right panel of Figure 16). This highlights the robust, plug-and-play nature of our framework, confirming its ability to generalize to scenarios with a greater number of subjects and motions.

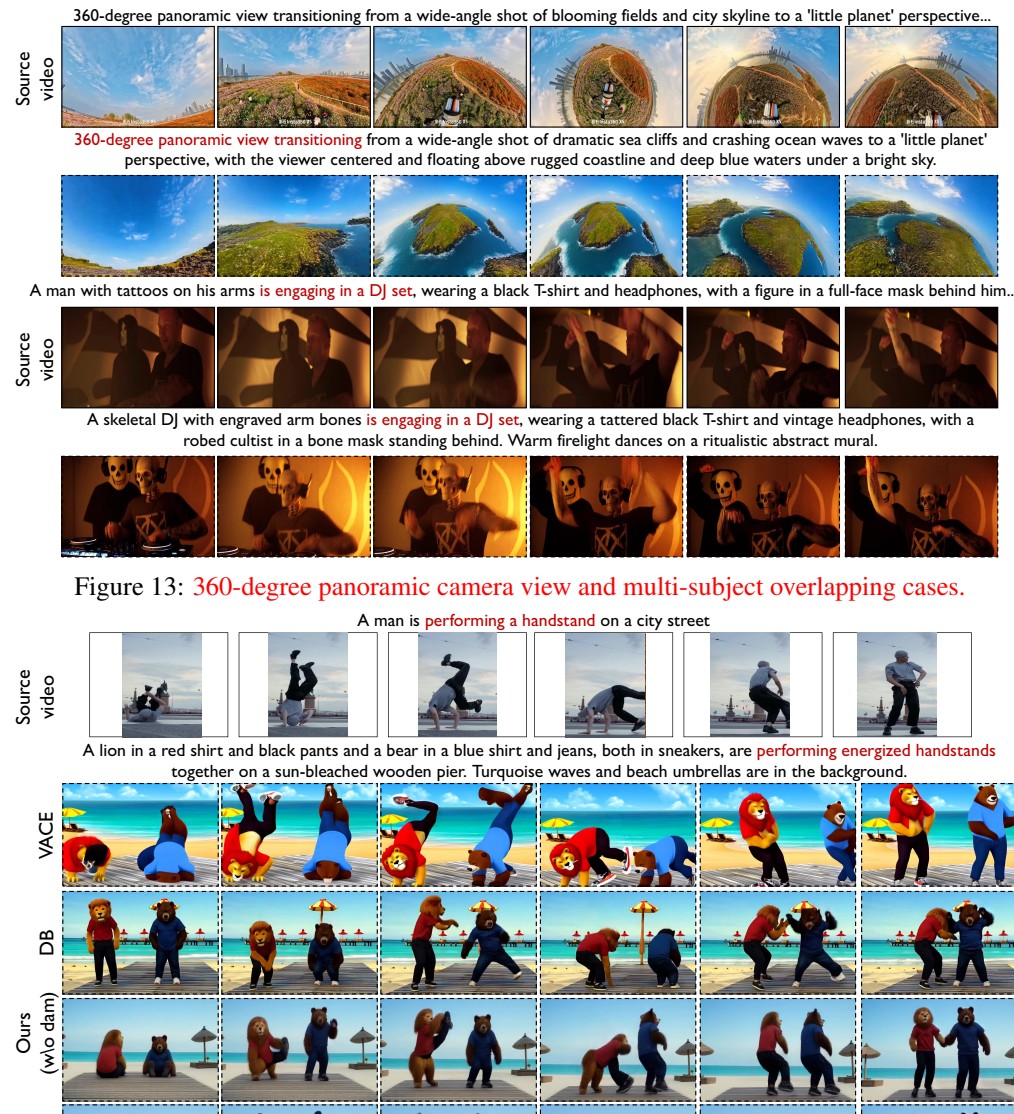

Figure 13: 360-degree panoramic camera view and multi-subject overlapping cases.

Figure 14: **More Compositional Motion Qualitative Comparison Results.** We compose the same motion in the generated video.

**New Metric for Multi-Motion Fidelity.** Existing motion fidelity metrics (Yatim et al., 2024; Shi et al., 2025) operate by computing a holistic similarity score between two sets of motion tracklets extracted from a reference and a generated video. However, this metric can not be used to evaluate the compositional motion setting for the motion blending effect. To compute the C&C score, we first construct the C&C similarities matrix $S^{CC}$, where each $ij$-th entry represents the motion similarity between cropped video $B_i$ and reference $V_{ref}^i$. Similarly, we compute the ground-truth similarities $S^{GT}$, where each $ij$-th entry represents the motion similarity between reference videos $V_{ref}^i$ and $V_{ref}^j$. We compute $||S^{CC} - S^{GT}||$ and get a matrix where diagonal entries denote motion similarity to the corresponding video, while off-diagonal entries are similarities to other reference videos, which represent the degree of motion blending. We provide additional evidence in Figure 17, where the $C\&C$ matrix, $C\&C$ score, and original motion fidelity are computed for every video. For successful videos, where every subject is assigned an accurate motion, the difference between $S^{CC}$ and $S^{GT}$ is small and results in high C&C scores. In cases where the motions corrupt each other and are incomplete and distorted, the difference becomes larger, and the C&C scores decreases.

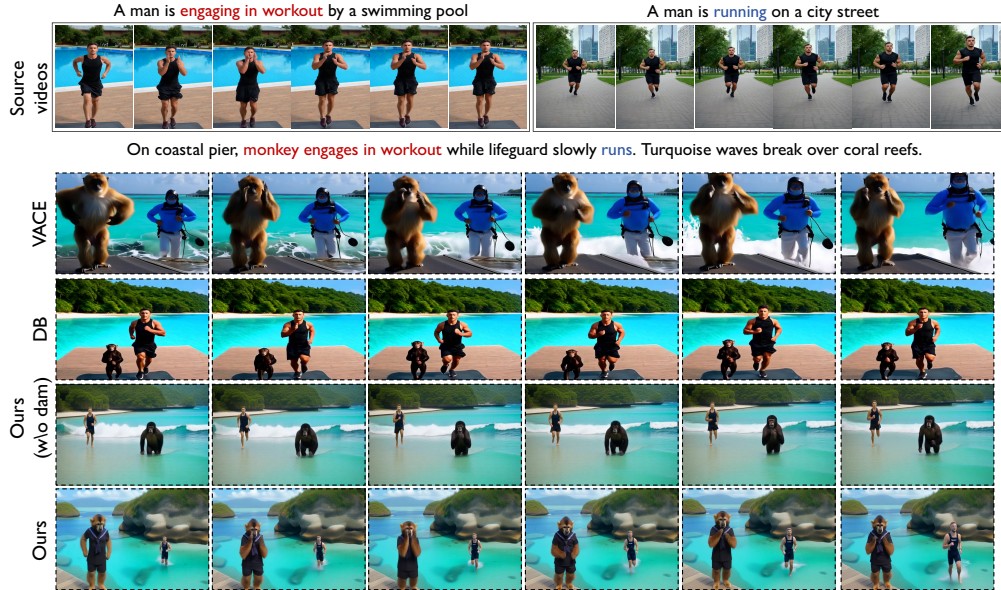

Figure 15: **More Compositional Motion Qualitative Comparison Results.** We compose the different motions in the generated video.

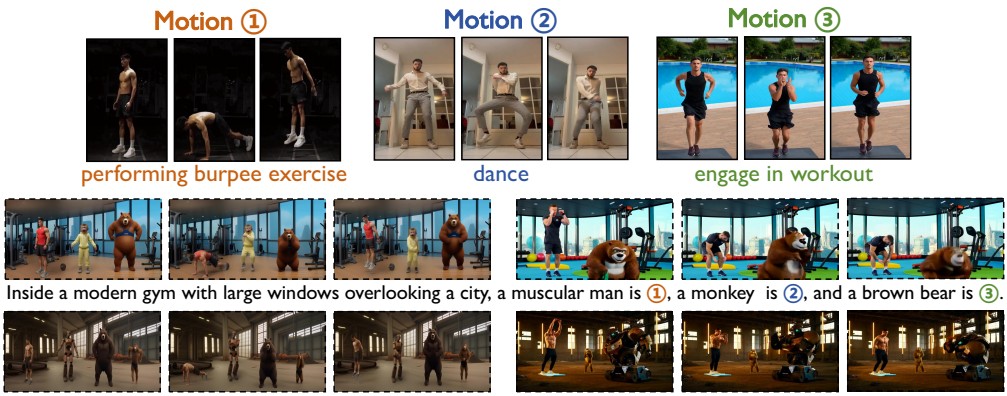

Figure 16: **Composition of more than two motions.** The first row shows the reference videos. The second row demonstrates a three-subject scene where each subject is assigned a distinct motion. The last row presents a four-subject composition: the bear and monkey perform the `engaging in a workout` motion, the human performs the `burpee` motion, and the robot is `dancing`. The left panel is generated by our method, while the right is generated by the base model using the same prompt.

However, due to the multi-motion blending, the blended videos get higher original motion fidelity scores. For example, in the 2nd and 3rd rows, an astronaut is assigned the first motion, and a woman is assigned the dance motion. In the successful videos, both the astronaut and the woman perform accurate motions, therefore getting a high C&C score, while for the blended one, the woman's motion is corrupted by the first motion, while the astronaut's motion is incomplete, leading to a significant difference between $S^{CC}$ and $S^{GT}$. However the original motion fidelity metric fails to distinguish between these cases effectively.

**Correlation with human evaluation.** Following the protocol of MuDI(Jang et al., 2024), we conducted a correlation study between human evaluation and the metrics ($C\&C$ score and original Motion Fidelity ). We generated 120 videos across four methods (DreamBooth, VACE, CoMo w/o DAM, and CoMo) and asked human raters to evaluate "Compositional Motion Fidelity" (Success/Fail). We calculated Spearman's

Table 3: Correlation between metrics and human evaluation.

| Metric | Spearman | AUROC |
|---|---|---|
| $C\&C$ Score | 0.50 | 0.71 |
| Motion Fidelity | 0.19 | 0.56 |

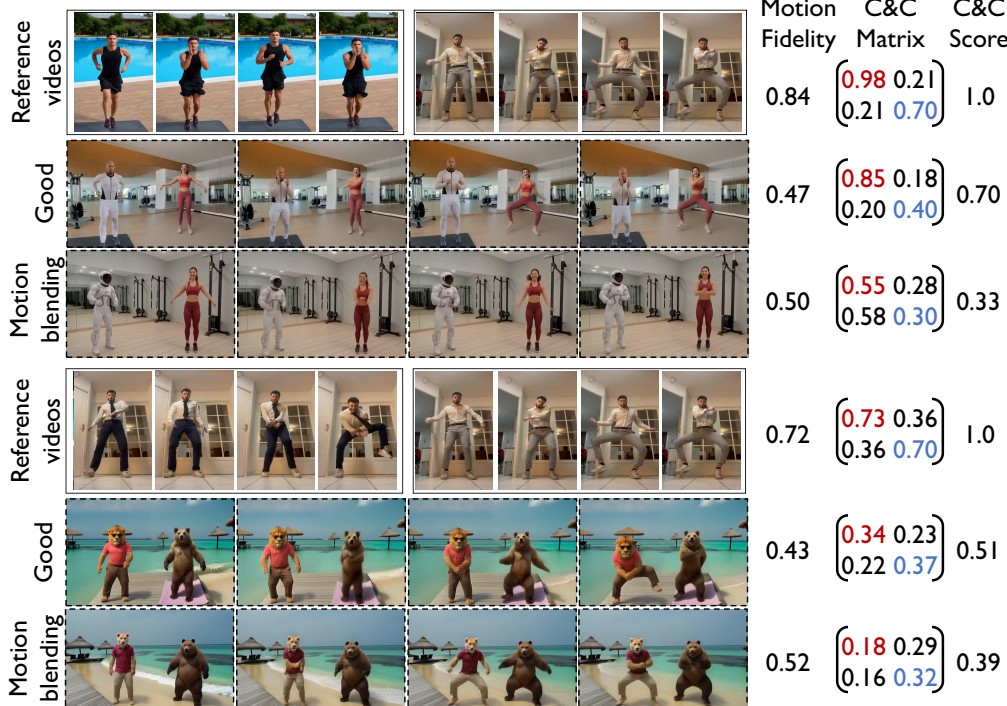

Figure 17: **Qualitative examples of the new matrix.** We visualize the motion fidelity and our C&C score for various cases. The C&C matrix of the reference videos corresponds to the gt matrix.

Table 5: Computational analysis in single motion setting.

|  | MotionDirector | DeT | DreamBooth | MotionClone | MOFT | Ours |
|---|---|---|---|---|---|---|
| Inference Time (s) | 168 | 440 | 150 | 804 | 576 | 150 |
| GPU Memory (G) | 31.1 | 30.9 | 26.5 | 51.5 | 75 | 31.5 |

rank correlation and AUROC between human scores and automatic metrics. As shown in the Table 3, our $C\&C$ score aligns significantly better with human perception than standard motion fidelity metrics, proving its reliability. Note that Spearman's rank correlation score was computed using the normalized sum of all human evaluation answers(e.g., if 3 out of 5 raters answered "Success" the score is 0.6). The AUROC was computed based on the majority voting results (0 or 1).

**Computational Efficiency Analysis.** We also analyze the inference time of our method with the increasing number of distinct motions. As shown in Table 4, the inference time

Table 4: Inference Time with more motions.

|  | Base Model | 2 Motions | 3 Motions | 4 Motions |
|---|---|---|---|---|
| Inference Time | 150s | 194s | 276s | 360s |

of our method increases only slightly as the number of motions grows. This is because, as the number of motions increases, the global latent is divided into smaller regional latents. Consequently, the computational cost for processing each individual region is significantly reduced. Although our method performs a forward pass for each motion, the reduced size of the input latents makes each pass substantially faster. The total computation, therefore, does not scale linearly with the number of motions, leading to only a marginal increase in total inference time.

**Compared with more baselines.** We also compared our method with state-of-the-art training-free single motion customization methods, including MOFT(Xiao et al., 2024) and MotionClone(Ling et al., 2024). As shown in Table 6, our method significantly outperforms these methods. Besides, these methods need additional guidance, which is very time-consuming and computational overhead, as shown in Table 5.

Table 6: Comparison with more existing methods.

| Method | Text Sim | Motion Fid | Temp. Cons. |
|---|---|---|---|
| Ours | 0.470 | 0.865 | 0.967 |
| MotionClone | 0.436 | 0.782 | 0.917 |
| MOFT | 0.437 | 0.540 | 0.966 |

# E    ADDITIONAL RELATED WORKS

**Text-to-Video (T2V) Generation.** Recent years have witnessed remarkable progress in T2V generation, largely propelled by advancements in diffusion models (Song & Ermon, 2019; Ho et al., 2020; Dhariwal & Nichol, 2021). Early methods (Wang et al., 2023; Zhao et al., 2024) often extended pre-trained T2I models to generate videos. They are originally based on U-Net (Ronneberger et al., 2015) architecture and incorporate additional temporal modules to model dynamics. While effective, these methods sometimes struggled with temporal consistency. A significant architectural shift towards DiTs (Peebles & Xie, 2023) has marked a new era for video generation. Leveraging their superior scalability and capacity for capturing long-range spatiotemporal dependencies, DiT-based models like Sora (Brooks et al., 2024), CogVideoX (Yang et al., 2024), and Wan (Wan et al., 2025) have shown state-of-the-art performance, generating high-fidelity and temporally coherent videos from complex textual prompts. However, despite their advance in generating diverse subjects and scenes, they struggle with precise motion control, particularly for complex, multi-subject motions.

**Compositional Visual Generation.** While compositional generation in the image domain has seen significant advancements, enabling the synthesis of multiple subjects (Kumari et al., 2023; Jang et al., 2024; Xie et al., 2023; Liu et al., 2022; Gal et al., 2022) or the combination of content and style (Xu et al., 2024; Shah et al., 2024; Wang et al., 2024), the composition of distinct motions in video remains a relatively underexplored frontier. To achieve this target, naively applying techniques from image composition, such as linearly merging motion-specific LoRAs (Ryu, 2023) often fails. This typically results in motion interference, where the dynamics blend together into an incoherent or corrupted result. Alternative approaches rely on external control signals like motion trajectories (Yin et al., 2023), object regions (Wu et al., 2024), and skeletal poses (Ma et al., 2024; Jiang et al., 2025; Xing et al., 2024; Zhang et al., 2023). However, these methods face several challenges. First, they often require training a signal encoder on large-scale datasets, demanding significant computational resources. Second, their performance is highly contingent on the quality of the extracted control signals. For example, in source videos with complex actions or occlusions, poorly extracted poses will degrade the fidelity of the generated motion. Furthermore, composing distinct motions from different source videos with these methods requires cumbersome preprocessing, which is not user-friendly. In this work, we propose a training-free divide-and-merge strategy that overcomes these limitations. Our method can flexibly assign specific motions to multiple subjects, pushing the boundaries of controllable video generation.

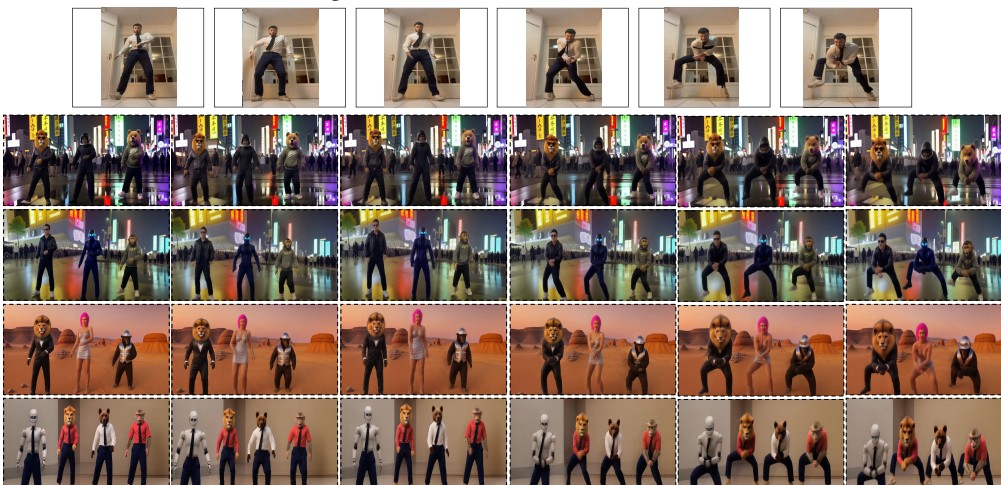

Figure 18: **More Compositional Motion Customization Results.** Each source video is combined with four newly generated videos. Here, we assign the motion in the source video to three or even four different subjects (last row) in the generated video.

# F    LIMITATIONS AND FUTURE WORK

Our method, while effective, has several limitations that present clear avenues for future research. First, the initial motion learning phase can occasionally introduce temporal flickering (see Fig-

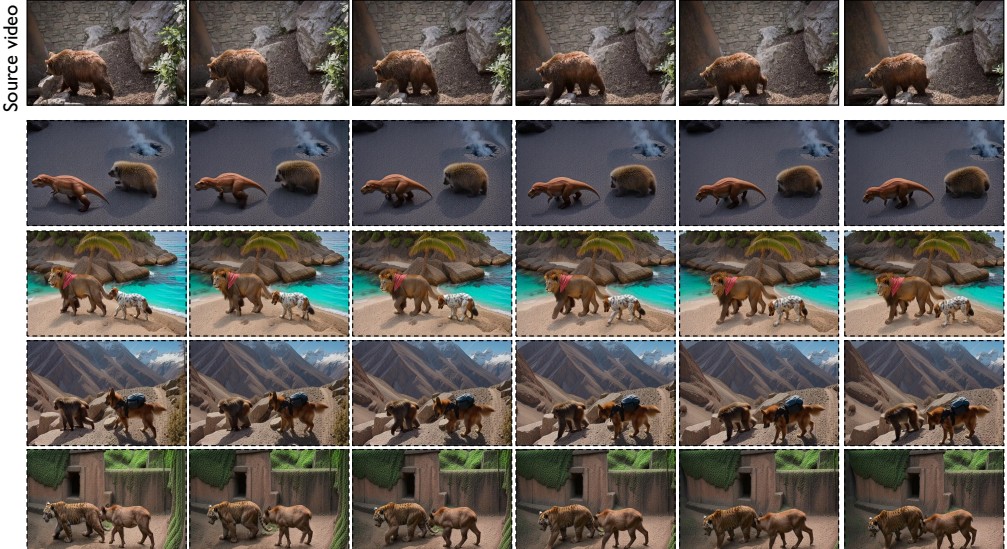

Figure 19: **More Compositional Motion Customization Results.** Each source video is combined with four newly generated videos. Here, we assign the motion in the source video to two different subjects in the generated video.

---

**Algorithm 1** Multi-Motion Composition

---

**Require:** Regions $r_{1,...,N}$, Target prompt $P_{tgt}$, motion-specific prompts $P_{tgt}^{1,...,N}$
    **Require:** LoRA sets $\Delta\theta_d^{1,...,N}$, guidance strengths $s, s_{1,...,N}$, weights matrix $w_i$
    **Require:** Global blending weight $\lambda$, diffusion steps $T$, global blending threshold $\tau$.
1:   $x_T \sim \mathcal{N}(0, \mathbb{I})$                                     ▷ Initialize from Gaussian noise
2:   $Z = \frac{1}{\sum_i w_i}$                                   ▷ Compute weight normalization
3:   $\overline{\Delta\theta_d} = \frac{1}{N}\sum_i \Delta\theta_d^i$                         ▷ Average LoRAs for global guidance
4: **for** $t \in [T, \ldots, 1]$ **do**                              ▷ Denoising steps
5:      **for** $i \in [1, \ldots, N]$ **do**                ▷ Compute regional velocity fields
6:          $\hat{v}_{r_i} = (1 + s_i)\, v_{(\theta+\Delta\theta_d^i)}\left(x_{t,r_i}, t, P_{tgt}^i\right) - v_{(\theta+\Delta\theta_d^i)}\left(x_{t,r_i}, t, \emptyset\right)$
7:      **end for**
8:      $\hat{v}_{local} = Z \odot \sum_{i=1}^N w_i \odot Pad_{r_i}\left(\hat{v}_{r_i}; x_t\right)$      ▷ Aggregate regional velocities
9:      **if** $t > \tau$ **then**             ▷ Apply global blending only during initial steps
10:        $\hat{v}_{global} = (1 + s)\, v_{(\theta+\overline{\Delta\theta_d})}\left(x_t, t, P_{tgt}\right) - v_{(\theta+\overline{\Delta\theta_d})}\left(x_t, t, \emptyset\right)$    ▷ Global prediction
11:        $\hat{v}_{final} = \lambda \cdot \hat{v}_{global} + (1 - \lambda) \cdot \hat{v}_{local}$      ▷ Linearly interpolate for final velocity
12:      **else**
13:        $\hat{v}_{final} = \hat{v}_{local}$              ▷ Use only local prediction in later steps
14:      **end if**
15:      $x_{t-1} = \text{Scheduler}(x_t, \hat{v}_{final})$                 ▷ Perform denoise step
16: **end for**
17: **return** $x_0$

---

ure 20), an artifact that degrades overall video quality and lowers temporal consistency scores. While this can be mitigated in a post-processing step using video restoration models (Wang et al., 2025), a more integrated solution that enforces temporal stability during training would be preferable. Second, the success of the multi-motion composition phase is highly contingent on the quality of the learned LoRA modules. An impure or entangled motion representation from the first phase is detrimental to the composition stage, often resulting in significant motion blending and distorted outputs. Finally, our divide-and-merge strategy is primarily designed for composing local, regional motions. It struggles to simultaneously handle global attributes, such as composing a specific camera motion with a local character's action. Future work should focus on designing a unified framework that can flexibly compose both global and local motion, thereby enabling more complex and dynamic video generation scenarios.

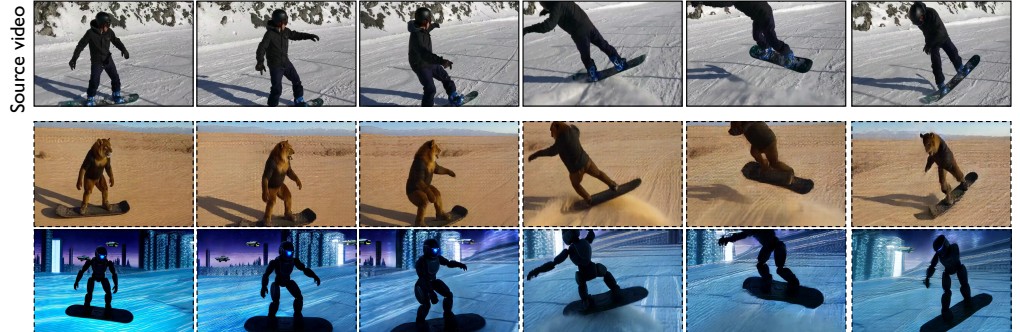

Figure 20: **Limitations.** The motion learning process in the first phase may cause temporal flickering.

## G  IMPLEMENTATION DETAIL

We utilized the Gemini-2.5 Pro (Comanici et al., 2025) as an LLM to serve as a writing assistant to polish the language of the manuscript.

## H  HANDLING SPATIAL OVERLAP AND INTERACTION

Our Divide-and-Merge strategy handles soft spatial overlaps d latents unit where d =32 in Figure 4 via Gaussian transitions, ensuring seamless visual blending at boundaries. However, there is an intrinsic trade-off: increasing the overlap width $d$ improves global spatial consistency (smoother transitions) but risks compromising motion integrity. A larger $d$ introduces more interference between the distinct motion LoRAs in the overlapping region, potentially diluting the specific motion patterns. We empirically selected $d = 32$ to balance this trade-off. For physical interaction: if the interaction exists within a single reference video (e.g., two people dancing together in Figure 13), our Phase 1 handles this effectively as a single motion pattern.

