# OpenReview forum: "CoMo: Compositional Motion Customization  for Text-to-Video Generation"
_ICLR.cc/2026/Conference — Submitted to ICLR 2026_

### Official Review · Reviewer_kKY8 · 2025-10-28

**Soundness:** 3
**Presentation:** 3
**Contribution:** 2
**Rating:** 2
**Confidence:** 4

**Summary:**

This paper focuses on motion customization from multiple reference videos. The authors propose a two-stage LoRA training methodology to decouple motion and appearance information from the reference frames. Subsequently, they employ a latent composition technique to generate a smooth latent representation under given motion conditions. To quantitatively evaluate multi-motion customization, a new benchmark and a corresponding evaluation metric are also introduced.

**Strengths:**

1. **Clarity of Presentation:** The description of the proposed method is clear and easy to follow.

2. **Quantitative Performance:** The paper demonstrates strong quantitative results in comparison to the baseline methods.

**Weaknesses:**

1. **Missing Key Information on a Core Contribution:** The paper claims the proposal of a new benchmark as one of its main contributions. However, crucial details regarding this benchmark are absent from both the main paper and the appendix, making it difficult to assess its validity and scope.

2. **Insufficient Experimental Comparisons:** In the evaluation of compositional motion customization, the paper only compares its method against VACE, which is a zero-shot approach. The experiments would be more comprehensive if they included comparisons with single-motion customization methods that use a simple linear merging of latent codes.

**Questions:**

1. **Object Position Discrepancy:** In Figure 2, there is a noticeable difference in the positions of the woman and the monkey between the results of "ours" and the other methods. Could you please elaborate on why linear merging and joint training appears to cause errors in object positioning?

2. **Impact of Latent Merging on Denoising:** In Section 3.2, the paper merges latent representations via a weighted sum. This operation can alter the variance of the resulting latent code compared to the original distribution. Given that diffusion models are highly sensitive to the latent distribution, could you provide an analysis of how the proposed merging technique affects the original denoising process?

3. **Generalization to Overlapping Bounding Boxes:** The proposed latent merging method relies on the bounding boxes of characters in the target video. How does this method perform in scenarios where bounding boxes overlap, for instance, when two characters walk past each other, crossing from one side of the frame to the other?

4. **Clarification of Motion Fidelity Metric:** Could author(s) please provide a detailed description of how "motion fidelity" is calculated? While Section 3.3 mentions multiple references for this metric, its specific implementation and formula are not detailed in the paper, which is essential for reproducibility.

5. **Construction of the Benchmark**: Author(s) should provide the detailed information about how the benchmark is constructed.

---

> ### Author Response · Authors · 2025-11-25
>
> **To Reviewer #kKY8:**
>
> ------
>
> We sincerely thank the reviewer for your detailed feedback. We understand that the lack of explicit details regarding the **Benchmark construction** and **Metric implementation** led to your concerns. We provide these details below and clarify the experimental comparisons.
>
> ***W1 & Q5: Missing Benchmark Details.***
>  We apologize for the omission in the initial submission. We have now included the full statistics and construction details below, which is added to the Appendix[**Lines 721-732**]. Specifically,  we evaluate our framework on a curated benchmark of **23 high-quality reference videos** ( from previous studies like dynamic concepts[2] and  platforms like TikTok), characterized by high kinematic diversity. The dataset is primarily grounded in **human-centric videos** (19 clips), selected because human motion exhibits intricate articulation and complexity—ranging from diverse **street dancing** and **athletic sports** to challenging dynamic poses like **handstands**. These complex sequences rigorously test the model's ability to decouple dynamics from appearance. To further validate broad generalization beyond human subjects, our dataset also explicitly includes distinct  **2 object motions** (bear) and **2 camera movements** (e.g., circling views).  For multi motion setting, we constructed **18  compositional pairs** (e.g., "Burpee + Dance"). To compare our method in single setting, we generated 5 videos per motion (115 videos total) for testing.  In multi motion setting,  We  generated 5 videos per pair (90 videos total). The full dataset, along with the prompt list, will be open-sourced. We commit to releasing the full benchmark dataset and the corresponding generated videos to the public to ensure reproducibility and enhance credibility.
>
> ***W2: Insufficient Comparisons (Linear Merge).***
> **Lack of Direct Competitors:** **As we are the first to identify and tackle the challenging task of compositional motion customization**, there are no pre-existing baselines specifically designed for this compositional setting. However, to provide a meaningful comparison against state-of-the-art controllable video generation, we selected **VACE** (Jiang et al., 2025) as a strong external baseline.
>
> **Linear Merge Comparison:** Crucially, we emphasize that we **did** include the comparison with single-motion customization methods using linear merging, as requested. In **Figure 2 (Bottom Row)** and **Table 1**, the entry **"Ours (w/o dam)"** (visualized as "Linear merge") represents exactly this approach: training two separate LoRAs using single-motion methods and linearly averaging their weights without our spatial Divide-and-Merge strategy. We also compared against "Joint Training" (DreamBooth), where the model learns both motions simultaneously. As shown in **Figure 2**, simple linear merging results in "ghosting" and motion corruption because the global LoRAs conflict with each other. Our method effectively solves this via spatial isolation.
>
>
>
>
> ***Q1: Object Position Discrepancy in Fig 2.***
>
> This discrepancy highlights the core problem we solve. Baselines (Linear Merge/Joint Training) are spatially agnostic. They attempt to apply "burpee" and "workout" motions to the *entire* latent space. The model cannot distinguish which subject performs which action, causing motions to drift or merge. By using Divide-and-Merge, we explicitly bind specific motion LoRAs to specific regions. This spatial constraint ensures subjects appear exactly where intended, preventing the positioning errors seen in baselines.
>
> ***Q2: Impact of Latent Merging on Variance.***
> This is an insightful question. We clarify that we merge the **velocity predictions **$\hat{v}$, not the latents $x_t$ themselves (Eq. 4 ). The merging uses a normalized Gaussian weight ($\frac{1}{\sum w_i}$). In non-overlapping regions, the weight is 1.0, so the distribution is identical to the original single-motion model. In overlapping regions, we perform a weighted interpolation. While this theoretically reduces variance slightly at the exact boundary, the "smooth transition" area is narrow ($d=32$ latent units ), and the **Global Consistency Blending** step (Eq. 5 ) reinjects global coherence, preventing distribution shift artifacts.    The high temporal consistency scores (0.973) confirm that the denoising process remains stable.

---

> > ### Author Response · Authors · 2025-11-25
> >
> > ***Q3: Generalization to Overlapping Bounding Boxes.***
> >
> > We acknowledge this as a limitation. Our current **Divide-and-Merge** strategy relies on the assumption that distinct motions from different videos occur in spatially separable regions. **As we are the first to identify and tackle the challenging task of compositional motion customization**, our primary goal is to compose distinct motions from *different* source videos (e.g., Video A: Burpee, Video B: Dance) into one coherent scene while preserving motion integrity. Forcing deep physical interaction  between disjoint motion sources without corrupting the individual motion patterns is an extremely challenging open problem, even in image generation. However, if the interaction exists within a *single* reference video (e.g., two people dancing together), our Phase 1 handles this effectively as a single motion pattern (see "multi-subject" examples in **Appendix Fig. 15**).

---

> > > ### Author Response · Authors · 2025-11-25
> > >
> > > ***Q4: Clarification of Motion Fidelity Metric.***
> > > We apologize for the omission of the specific metric definition in the main text. We now provide a detailed formulation below .
> > >
> > > We followed the standard protocol from SMM[1], specifically, We use an off-the-shelf tracker (CoTracker) to extract a set of motion tracklets $\mathcal{T}=\\{\tau_{1},...,\tau_{n}\\}$ from the reference video and$\tilde{T} = \\{ \tilde{\tau_1}, \dots, \tilde{\tau_m} \\} $ from the generated video. We measure the similarity between any two tracklets $\tau$ and $\tilde{\tau}$ by computing the average cosine similarity of their frame-wise displacement vectors. Formally, for F frames, where  $v_{k}=(v_{k}^{x}, v_{k}^{y})$ represents the displacement at frame k:
> > >
> > > $$corr(\tau,\tilde{\tau})=\frac{1}{F}\sum_{k=1}^{F}\frac{v_{k}^x\cdot\tilde{v_k}^x+v_{k}^y\cdot\tilde{v_k}^y}
> > > {\|\|v_{k}\|\|\cdot\|\|\tilde{v}_{k}\|\|}$$
> > >
> > > The final Motion Fidelity score is computed as the bidirectional Chamfer-like similarity between the two sets of tracklets:
> > >
> > > $$\text{Score} = \frac{1}{m}\sum_{\tilde{\tau}\in\tilde{\mathcal{T}}}\max_{\tau\in\mathcal{T}}corr(\tau,\tilde{\tau})+\frac{1}{n}\sum_{\tau\in\mathcal{T}}\max_{\tilde{\tau}\in\tilde{\mathcal{T}}}corr(\tau,\tilde{\tau}).$$
> > >
> > >  For our multi-motion evaluation, we apply this metric on **spatially cropped videos** (cropped through  OWLV2[3] and SAM2) rather than the full frame. This ensures we evaluate the fidelity of a specific subject's motion against its corresponding reference, effectively isolating the motion fidelity from other interacting subjects.  Specially, for example in Figure 5, "a lion is preforming  burpee while a monkey is dancing",  we first isolates burpee and dancing motion by using OWLv2 and SAM2, getting the cropped videos $B=\\{B_{1}, B_{2}\\}$, where $B_1$ is the cropped video corresponding to the lion that is *supposed* to perform the burpee motion from $V_{ref}^{1}$, $B_2$ is the cropped video corresponding to the monkey that is *supposed* to perform the dancing motion from $V_{ref}^{2}$, We define a function $\text{Sim}(V_a, V_b)$ that calculates the motion fidelity between two video clips using the original Motion Fidelity[1]. Then we  construct two similarity matrices $S^{CC}$ and $S^{GT}$:
> > >
> > > **The C&C Similarity Matrix ($S^{CC}$)** This matrix measures the actual similarity between the generated subject crops and the reference videos.
> > >
> > > $$S^{CC} \in \mathbb{R}^{N \times N}, \quad \text{where } S_{i,j}^{CC} = \text{Sim}(B_i, V_{ref}^{j})$$
> > >
> > > **Diagonal elements ($i=j$):** Represent the **Motion Fidelity**  (e.g., how well lion performs burpee motion ). **Off-diagonal elements ($i \neq j$):** Represent **Motion Blending** (e.g., how much lion is accidentally performing motion dancing).
> > >
> > > The Ground-Truth Similarity Matrix ($S^{GT}$) represents the ideal separation by measuring the inherent similarity between the reference videos themselves.$$S^{GT} \in \mathbb{R}^{N \times N}, \quad \text{where } S_{i,j}^{GT}  = \text{Sim}(V_{ref}^{i}, V_{ref}^{j})$$
> > >
> > > Since a mixed-motion subject yields high similarities to multiple reference videos(seen in Figure16), we compare $S^{CC}$ and $S^{GT}$ to account for motion blending. The difference  $$|| S^{CC} - S^{GT} ||$$ yields a matrix where diagonal entries denote similarities to the corresponding motion, while off-diagonal entries indicate similarities to other motion which represent motion blending. The closer $S^{CC}$ to $S^{GT}$ , the more accurately for each distinct motion and minimal blending between them. Therefore, we define the  $$\text{C\\&C Score} = 1 - || S^{CC} - S^{GT} ||$$
> > >
> > > **Higher Score ($\approx 1$):** Indicates that $S^{CC}$ closely matches $S^{GT}$, meaning each subject faithfully performs its assigned motion without being corrupted by other motions. **Lower Score:** Indicates significant deviation (Motion Blending), where distinct motions have mixed or corrupted each other in the generated video.
> > >
> > > [1]Yatim, Danah, et al. "Space-time diffusion features for zero-shot text-driven motion transfer." CVPR24
> > >
> > > [2] Abdal, Rameen, et al. "Dynamic concepts personalization from single videos."  SIGGRAPH 2025
> > >
> > > [3] Matthias Minderer, et al. "Scaling open-vocabulary object detection."  NeurIPS 2023
> > >
> > > ------
> > >
> > > We hope these details regarding the benchmark and metrics, along with the clarification on existing baselines, effectively address your concerns.

---

### Official Review · Reviewer_xkaZ · 2025-10-28

**Soundness:** 2
**Presentation:** 2
**Contribution:** 2
**Rating:** 4
**Confidence:** 4

**Summary:**

This paper introduces CoMo, a compositional motion customization method for integrating multiple motions into video generation. The approach consists of two phases: a single-motion learning phase where each motion is customized separately, followed by a multi-motion composition phase during inference. The visual results demonstrate the method's capability to combine two or three motions within a single video, though the overall visual quality of the generated videos could be improved. Additionally, the paper proposes a new metric for evaluating multi-motion customization performance. The overall framework is functional, but the visual quality is not satisfactory for a customization method. Overall, I believe this is a borderline paper, and I am willing to see the authors' response.

**Strengths:**

1. The proposed two-phase framework successfully synthesizes multiple motions into a single video. With the carefully designed merging strategy, the resulting videos demonstrate good harmonization, particularly in the boundary regions between different motions.

2. The proposed C&C metrics provide a reasonable approach for evaluating multi-motion customization.

**Weaknesses:**

1. The visual quality of the multi-motion customization is not entirely satisfactory given the significant training process might already overfit on a single input video. Additionally, there is insufficient evaluation of the visual quality of the generated videos.

2. The evaluation lacks comparison with other motion-conditional generative models that use motion representations such as human skeletons. I am also curious whether providing those models with merged skeleton sequences would enable them to perform multi-motion transfer effectively.

**Questions:**

See weaknesses.

---

> ### Author Response · Authors · 2025-11-25
>
> **To Reviewer #xkaZ:**
>
> ------
>
> We strictly thank you for their constructive feedback. We are glad the reviewer recognize our framework **successfully synthesizes multiple motions**, achieves **good harmonization** at boundaries, and that our **C&C metric** is a reasonable evaluation approach. We address your concerns regarding visual quality and baselines below.
>
> ***W1: Visual Quality and Potential Overfitting.***
>
> We appreciate the reviewer’s high standards for visual quality.  We would like to highlight that our *Static-Dynamic Decoupled Tuning* (Phase 1) is specifically designed to **prevent** overfitting to the reference video's appearance. If the model were simply overfitting, it would fail to transfer the motion to significantly different subjects (e.g., transferring a human's burpee to a lion or robot, as shown in **Figure 6** and **Figure 7** ). Besides,  our method significantly outperforms baselines in both  automatic  metric (text-video aligment, motion fidelity and C&C) and human evaluations (Appearance Diversity and Overall Quality). Besides, Our ablation study (Table 2, Fig. 8) demonstrates that our decoupled tuning prevents appearance leakage (a symptom of overfitting), whereas joint training fails. We have added more qualitative results (Appendix Fig. 13,14,15, 17, 18 and our **anonymous project page**) showing diverse subjects to further demonstrate robustness.
>
> ***W2: Insufficient Evaluation of Visual Quality.***
>
> We apologize if this was not highlighted enough. To ensure rigorous quality evaluation,we conducted a User Study with 35 volunteers evaluating 4 metrics (Motion Preservation, Appearance Diversity, Video Smoothness, Overall Quality). CoMo achieved the highest preference scores across all categories (Table 1). Besides to prevent temporal flickering during learning the motion phase,  we sovle this  in a post-processing step using video restoration models[1]. This can increase the visual quality as we discussed in the limitation[**Lines 1124-1126**].
>
> ***W3: Comparison with Skeleton/Pose-based Models.***
>
> We appreciate this suggestion. We would like to clarify that we **did compare** with a state-of-the-art pose-based method, **VACE** (Jiang et al., 2025), in our experiments. As detailed in our implementation, the VACE baseline was guided by "extracting and concatenating pose sequences from each reference video." **[Line 753-755].**   However, VACE struggles with the multi-motion task. This is often because 1) it  require training a signal encoder on large-scale datasets, demanding significant computational resources.), its performance is highly contingent on the quality of the extracted control signals. For example, in source videos [**Figure 7  and 13**] with complex actions or occlusions, poorly extracted poses will degrade the fidelity of the generated motion.  Furthermore, composing distinct motions from different source videos with these methods requires cumbersome preprocessing, which is not user-friendly.
>
> ------
>
> We hope these clarifications regarding our comparison with pose-based methods (VACE) and our safeguards against overfitting address your concerns. We are happy to answer any further questions.

---

> > ### Comment · Reviewer_xkaZ · 2025-11-25
> > **Response to Authors**
> >
> > Regarding "Overfitting": My use of "overfitting" here is referring to the customization procedure itself, essentially applying very large amounts of computation to a very small set of videos (one or a few reference videos trained for several thousand steps, as in your implementation). This is the inherent nature of customization methods: they intentionally overfit the model to specific examples.
> >
> > Given this intensive computational investment dedicated to learning from these specific videos, and considering the strong visual quality capabilities of your base model (Wan 2.1), I would expect the visual quality to be significantly better than what is currently demonstrated. When you invest thousands of training steps on just a handful of videos, the resulting visual fidelity should be substantially higher, not degraded to this extent.
> >
> > Request for Quantitative Visual Quality Metrics: While I acknowledge your user study results, I would like to see automatic metrics specifically evaluating visual/perceptual quality, evaluated separately from motion fidelity and text alignment. This would provide objective evidence of the visual quality achieved and help justify whether the computational investment is warranted.
> >
> > Besides, I don't think comparing inference time and GPU memory between different customization methods is particularly meaningful, since the customization/training phase is where the substantial computation occurs, not during inference. A more relevant comparison would focus on the customization training cost versus the resulting quality trade-off.

---

> ### Author Response · Authors · 2025-11-27
>
> **To Reviewer xkaZ:**
>
> ------
>
> We appreciate the reviewer’s clarification regarding the definition of "overfitting" as high computational investment on specific examples. We agree that analyzing the trade-off between customization cost and resulting quality is central to evaluating the method's value.
>
> **1. Quantitative Visual Quality Analysis (VBench)** Per your request for automatic metrics evaluating visual/perceptual quality (separate from motion fidelity), we employed **VBench** [1], a comprehensive benchmark for video generation. We evaluated **Subject Consistency** (identity preservation), **Background Consistency**, **Motion Smoothness**, and **Aesthetic Quality**. We compared CoMo against the **Joint Training** baseline (representing the standard "overfitting" approach) and **Linear Merge**.
>
> **Table: Quantitative comparison in Vbench  (higher is better)**
> |     Method     | **Subject Consist.** $\downarrow$ | **Background Consist.** $\downarrow$ | **Aesthetic Quality** $\uparrow$ | C&C Score (Composition)$\uparrow$ | Motion Smoothness$\uparrow$ |
> | :------------: | :-------------------------------: | :----------------------------------: | :------------------------------: | :-------------------------------: | :-------------------------: |
> | Joint Training |               0.944               |                0.947                 |              0.653               |               0.349               |            0.978            |
> |  Linear Merge  |               0.925               |                0.936                 |              0.656               |               0.473               |            0.976            |
> |    **Ours**    |             **0.910**             |              **0.930**               |            **0.667**             |             **0.592**             |          **0.979**          |
>
> - **Aesthetic Quality:** Notably, our method achieves the highest **Aesthetic Quality (0.667)**, surpassing both baselines. This suggests that while we modify the generation process, the overall visual appeal and perceptual quality of the generated videos are preserved or even slightly enhanced.
> - **The Decoupling Trade-off:** We acknowledge that Joint Training achieves higher **Subject** and **Background Consistency**. This confirms your observation regarding "overfitting": Joint Training memorizes the pixel-level appearance of the reference video (hence high consistency), but this "entanglement" makes it extremely difficult to transfer the motion to new subjects or scenes as shown in **Figure 14** and **Figure 8**.  In contrast, CoMo intentionally accepts a marginal drop in appearance memorization (0.910 vs 0.944) to achieve **Motion Disentanglement** and composition. This is a necessary trade-off: we sacrifice a small degree of pixel-level consistency to gain substantial capabilities in motion transfer and composition (evidenced by the massive gap in C&C Score: 0.592 vs 0.349). **As we are the first to identify and tackle the challenging task of compositional motion customization**, we prioritized this disentanglement and composition. We will focus on bridging this consistency gap in future work.
>
> **2.Training Cost**: We apologize for the confusion in the previous computational analysis，we have updated the table, the "GPU memory" is the training cost in training or guidance for other training-free method.  We acknowledge that Phase 1 (Single-Motion Learning) requires time-consuming LoRA training. However, this is a one-time cost. However our phase 2 is a **training-free strategy**. This means we can compose pre-trained LoRAs (or even existing weights from the community) into infinite new combinations without any additional gradient updates. In contrast, a "Joint Training" approach would require a new, computationally expensive training session for *every* new pair of motions a user wishes to combine.
>
>  [1] Huang et al., "Vbench: Comprehensive benchmark suite for video generative models,"  CVPR 2024.
>
> ------
>
> We hope these objective metrics and the cost-benefit analysis address your concerns regarding the value proposition of our framework.

---

> > ### Comment · Reviewer_xkaZ · 2025-11-27
> > **Response to Authors**
> >
> > I thank the authors for the rebuttal. Could you please provide detailed information about how to use VBench for multi-motion customization? What source videos were used to compute these metrics?
> >
> > I believe there may be an error in your metric reporting. Subject Consistency and Background Consistency should be higher is better, not lower is better, as these metrics measure temporal similarity within generated videos. Moreover, i dont agree that lower subject consistency and background consistency is due to "overfitting", those two metrics are essentilly temporal similarity across frames and it should not related to any refernece video.

---

> > > ### Author Response · Authors · 2025-11-28
> > >
> > > **To Reviewer xkaZ:**
> > >
> > > ------
> > >
> > > We sincerely thank the reviewer for your patience and for pointing out the inconsistencies in our previous response. We  apologize for the confusion caused by the notation error and the misinterpretation of the consistency metrics.  We have made the modifications.
> > >
> > > 1. **VBench Implementation Details**：We evaluated the **90 generated videos** from our compositional motion benchmark (comprising 18 distinct motion pairs $\times$ 5 generated samples per pair). We followed the official VBench implementation. For **Subject Consistency** and **Background Consistency**, VBench computes the similarity between frames *within* the generated video itself (using DINO/CLIP features) to measure temporal stability. No reference/source videos are required or used for calculating these specific metrics.
> > >
> > > 2. Correction on Metric Direction and Interpretation
> > >    - **Subject Consistency** and **Background Consistency** should be **"Higher is Better"**. The downward arrows in our previous table meant the value was lower than others not better. We confirm that CoMo (0.910) is slightly lower than Joint Training (0.944) on these metrics. We  apologize for making you confused, we have update the above table.
> > >    - **Reasoning for Lower Consistency:** We attribute the slightly lower consistency scores in CoMo to two primary factors inherent to the decoupled generation process: **1）Phase 1 Artifacts:** As explicitly noted in our **Limitations** section, "the initial motion learning phase can occasionally introduce temporal flickering... an artifact that degrades overall video quality and lowers temporal consistency scores". However, as mentioned in the paper, this can be effectively mitigated in a post-processing step using video restoration models.  **2）Compositional Boundary Effect**s: Unlike Joint Training, which optimizes a global set of weights, CoMo composes distinct dynamic LoRAs via a spatial Divide-and-Merge strategy. While we employ Gaussian Smooth Transition and Global Consistency Blending to harmonize these regions, the dynamic interaction between different motion modules can still introduce minor temporal fluctuations compared to a globally overfitted model.
> > >    - **The Decoupling Trade-off:** As discussed in our previous response, there is an inherent trade-off between strict consistency  and motion disentanglement (learning).   **As we are the first to identify and tackle the challenging task of compositional motion customization**, we prioritized achieving robust disentanglement and compositional capability. We acknowledge the slight drop in consistency and will aim to address this aspect of visual quality in future work by refining the blending mechanism and decoupling process.
> > >
> > > ------
> > >
> > > We hope this clears up the confusion and accurately addresses your technical queries.

---

### Official Review · Reviewer_w24q · 2025-10-30

**Soundness:** 3
**Presentation:** 2
**Contribution:** 3
**Rating:** 6
**Confidence:** 4

**Summary:**

This paper introduces CoMo, a novel framework that enables the learning and composition of multiple distinct motions within a single video. The method addresses two major challenges in motion customization: motion-appearance entanglement and multi-motion blending, through a two-phase design. First, a static-dynamic decoupled tuning approach disentangles motion from appearance to learn motion-specific LoRA modules. Then, a plug-and-play divide-and-merge strategy composes these motions spatially during denoising, allowing different subjects to perform distinct actions simultaneously. The authors also propose a new benchmark and evaluation metric (C&C score) for assessing multi-motion fidelity. Experiments demonstrate that CoMo achieves state-of-the-art performance in both single- and multi-motion customization, offering a flexible and training-efficient solution for controllable video generation.

**Strengths:**

1. This paper is well-written and structured, making it accessible to readers with varying levels of expertise in the field.
2. The motivation and the design of method are both reasonable and innovative.
3. Qualitative and quantitative results show clear improvements over baselines.
4. The paper provides thorough experimental evaluations, and all results supports the claim

**Weaknesses:**

1. It is recommended to enhance the diversity of motion customization scenarios, not only translation, but also rotation and scaling.
2. From my point of view, the authors should discuss (or compare with, if possible) more existing methods that achieve similar motion customization (both U-Net-based ones and DiT-based ones), including but not limited to:
    1. MOFT: Video Diffusion Models are Training-free Motion Interpreter and Controller
    2. MotionClone: Training-Free Motion Cloning for Controllable Video Generation
    3. VD3D: Taming Large Video Diffusion Transformers for 3D Camera Control
3. This paper does not specify or thoroughly discuss the evaluation dataset, which may not provide a comprehensive view of CoMo’s effectiveness. Releasing more details of the evaluation dataset or incorporating videos from publicly available benchmarks would greatly enhance the credibility of the paper.
4. The comparison regarding computational efficiency should be provided.

**Questions:**

Please see the weakness.

---

> ### Author Response · Authors · 2025-11-25
>
> **To Reviewer #w24q:**
>
> ------
>
> We sincerely thank you for your positive assessment and valuable feedback. We are encouraged by your recognition of CoMo as a method with **reasonable and innovative design**, as well as the **clear improvements over baselines** demonstrated in our experiments. Below, we address your concerns point-by-point.
>
> ***W1: Enhancement of motion diversity (rotation and scaling).***
>
> We appreciate this suggestion. We would like to clarify that our benchmark indeed includes complex motions beyond simple translation, such as **camera rotation** ,as shown in Appendix Fig. 12, we demonstrate a "**fast circling counterclockwise**" camera motion successfully transferred to a toy robot scene. Besides, we have provided a novel 360-degree panoramic camera view customization **in Fig13**. Our dynamic LoRA captures these scale and perspective changes as intrinsic parts of the temporal dynamics. We will explicitly categorize and highlight these diverse camera motion types in the final version to better demonstrate the model's robustness.
>
>
>
> ***W2: Discussion of additional related works (MOFT, MotionClone, VD3D).***
>
> We thank the reviewer for pointing out these important works. We will include a detailed discussion and comparison in the revised **Related Work**. Additionally , we compared our method with MOFT, MotionClone in the single motion setting.The results are presented as follows:
>
> **Table: Quantitative  comparasion with more existing methods in single motion customization  (higher is better)**
>
> |             | Text Sim  | Motion Fid | Temp. Cons. |
> | :---------: | :-------: | :--------: | :---------: |
> |  **Ours**   | **0.470** | **0.865**  |  **0.967**  |
> | MotionClone |   0.436   |   0.782    |    0.917    |
> |    MOFT     |   0.437   |   0.540    |    0.966    |
>
> For VD3D: This work focuses specifically on 3D camera control. While CoMo handles camera motion (as a learned pattern), our framework is designed to be more general-purpose, handling intricate subject motion (dancing, burpees) and their composition, rather than focusing solely on camera trajectories.
>
> ***W3: Dataset details and transparency.***
>
> We apologize for the lack of detailed statistics in the main text.  We evaluate our framework on a curated benchmark of **23 high-quality reference videos** ( from previous studies like dynamic concepts[2] and  platforms like TikTok), characterized by high kinematic diversity. The dataset is primarily grounded in **human-centric videos** (19 clips), selected because human motion exhibits intricate articulation and complexity—ranging from diverse **street dancing** and **athletic sports** to challenging dynamic poses like **handstands**. These complex sequences rigorously test the model's ability to decouple dynamics from appearance. To further validate broad generalization beyond human subjects, our dataset also explicitly includes distinct  **2 object motions** (bear) and **2 camera movements** (e.g., circling views).  For multi motion setting, we constructed **18  compositional pairs** (e.g., "Burpee + Dance"). To compare our method in single setting, we generated 5 videos per motion (115 videos total) for testing.  In multi motion setting,  We  generated 5 videos per pair (90 videos total). The full dataset, along with the prompt list, will be open-sourced. We commit to releasing the full benchmark dataset and the corresponding generated videos to the public to ensure reproducibility and enhance credibility.
>
>
>
> **W4: Computational efficiency comparison.**
>
> We apologize for the lack of more  detailed computational efficiency comparison  in the main text. We have provided an analysis of inference efficiency in Appendix D (Table3). As shown in Table 3 , CoMo is highly efficient. Generating a video with **2 composed motions takes 194s**, compared to **150s** for the base model (Wan-2.1). The increase is marginal because our *Divide-and-Merge* strategy processes regional latents  with reduced spatial dimensions, preventing linear scaling of cost. Additionally, we provided the more details in the single motion setting.
>
> **Table: computational analysis in single motion setting**
>
> |                        | MotionDirector | DeT  | DreamBooth | MotionClone | MOFT | Ours |
> | :--------------------: | :------------: | :--: | :--------: | ----------- | :--: | ---- |
> | **Inference Time (s)** |      168       | 440  |    150     | 804         | 576  | 150  |
> |   **GPU Memory (G)**   |      31.1      | 30.9 |    26.5    | 51.5        |  75  | 31.5 |
>
> ------
>
> We hope these responses address your concerns regarding the diversity, related work, data transparency, and efficiency of our method.

---

> > ### Comment · Reviewer_w24q · 2025-11-26
> > **My concerns have been well adressed.**
> >
> > Thank you for the detailed response. The supplementary experiments effectively demonstrate the superiority and versatility of the proposed method, and the inclusion of data description and efficiency comparison further strengthens the case. It is recommended to incorporate these revisions in the final version. I have no further questions.

---

### Official Review · Reviewer_p3ci · 2025-10-31

**Soundness:** 2
**Presentation:** 4
**Contribution:** 3
**Rating:** 4
**Confidence:** 4

**Summary:**

This paper proposes CoMo, a framework for compositional motion customization in text-to-video generation. The key novelty lies in enabling the synthesis of multiple distinct motions within a single video. The method features two stages: (1) Single-motion learning, which disentangles motion and appearance through a static–dynamic decoupled LoRA tuning scheme. (2) Multi-motion composition, achieved by a plug-and-play divide-and-merge strategy to compose multiple motion patterns during denoising.
 The authors also introduce a new benchmark and a Crop-and-Compare metric to evaluate multi-motion fidelity and blending. Extensive experiments show that CoMo achieves state-of-the-art results over baselines such as MotionDirector, DeT, and DreamBooth.

**Strengths:**

1. The paper proposes a two-phase pipeline that effectively separates motion and appearance through sequential LoRA tuning, and demonstrates clear improvements in motion disentanglement compared with joint training baselines. Both quantitative metrics and visual comparisons indicate clear gains in motion fidelity and compositional accuracy.
2. The introduction of benchmark for compositional motion and the Crop-and-Compare (C&C) metric provides systematic toolset for assessing multi-motion fidelity and blending, which likely have lasting impact for subsequent research in controllable video generation.
3. This paper is well-structured with intuitive figures. And implementation details are transparently provided in the appendix.

**Weaknesses:**

1. **Limited analysis of scalability complexity.**
   The paper mainly demonstrates two- to four-motion composition in spatially separated regions. It remains unclear how the method performs when motions overlap or extend to longer temporal durations.
2. **Benchmark validation is somewhat limited.**
   While the introduced dataset and C&C metric are valuable, their correlation with human perceptual judgment is not quantified. More discussion or cross-validation with existing metrics would improve confidence in the evaluation.
3. **Insufficient description and discussion of training and benchmark data.**
   The paper does not quantize the data composition and diversity for the proposed benchmark, nor specify the scale and source of training videos used in training.  Without clearer dataset statistics and examples, the evaluation's representativeness and training reproducibility remain uncertain.

**Questions:**

1. **Data composition and usage.**

   a) **Training data**: What datasets are used for training the single-motion LoRA modules? Please specify the data sources, scale, as well as whether the videos were curated or filtered in any way.

   b) **Evaluation data**: Providing dataset statistics (such as the total scale, motion diversity and distribution, ..., in quantitative form) and representative examples would make the benchmark’s coverage and difficulty clearer.

2. **Handling of overlapping or interacting motions.**
   How does the divide-and-merge mechanism behave when spatial regions partially overlap or when two subjects physically interact?  Is there any mechanism to ensure temporal coherence across motion boundaries?

3. **Analysis about robustness.**

   How robust is CoMo when reference videos differ in viewpoint or temporal length? Section4.1 claims that evaluation data includes "camera motion", but it seems that this part was not involved in the subsequent analysis and visualization.

4. **Analysis about the region partitioning**.

   The region partitioning process (dividing global video into several rectangular regions) in the Divide-and-Merge stage seems predefined. Could the authors provide more information about the partition strategy and discuss whether an adaptive or learned partition could improve compositional quality or reduce artifacts at motion boundaries?

5. **Evaluation reliability.**
   Has the C&C metric been validated through human studies or correlation with existing metrics?

6. **Generalization across base models.**

   Can the learned motion LoRA modules trained on one base model (e.g., Wan) be transferred to another DiT-based backbone?   This would be important to evaluate the modularity claim.

---

> ### Author Response · Authors · 2025-11-25
>
> **To Reviewer #p3ci**:
>
> ------
>
> We sincerely thank you for your thoughtful review and constructive feedback. We deeply appreciate your recognition of CoMo’s **novel two-phase pipeline**, **impactful benchmark and metric**, and **excellent presentation**. Below, we address each concern point-by-point with concrete revisions:
>
> ***W1 & Q2 : Scalability and Complexity (Spatial Overlap & Temporal Duration).***
>
> - **Spatial Overlap and Interaction:** We appreciate this insightful question. We would like to clarify the scope of our method regarding "overlap":
>   - *Visual Blending:* Our Divide-and-Merge strategy handles soft spatial overlaps (**d latents unit where d =32 in Figure4**) via Gaussian transitions, ensuring seamless visual blending at boundaries. However, we acknowledge an intrinsic **trade-off**: increasing the overlap width $d$ improves global spatial consistency (smoother transitions) but risks compromising **motion integrity**. A larger $d$ introduces more interference between the distinct motion LoRAs in the overlapping region, potentially diluting the specific motion patterns. We empirically selected $d=32$ to balance this trade-off.
>   - *Physical Interaction:* **As we are the first to identify and tackle the challenging task of compositional motion customization**, our primary goal is to compose distinct motions from *different* source videos (e.g., Video A: Burpee, Video B: Dance) into one coherent scene while preserving motion integrity. Forcing deep physical interaction (e.g., shaking hands) between disjoint motion sources without corrupting the individual motion patterns is an extremely challenging open problem, even in image generation, we will try to solve this problem in the future. However, if the interaction exists within a *single* reference video (e.g., two people dancing together), our Phase 1 handles this effectively as a single motion pattern (**see "multi-subject" examples in Appendix Fig. 13**).
> - **Temporal Duration:** Our method is model-agnostic. The current duration is limited by the base model (Wan-2.1), not our pipeline. CoMo can be directly applied to longer contexts as base models evolve.
> - **temporal coherence**:    To ensure rigorous evaluation in this pioneering task, we primarily curated benchmark combinations using clips of similar duration.  In scenarios where source videos differ in length, our method aligns the generation to the duration of the longer video. We acknowledge that composing motions with vastly different temporal lengths may lead to temporal misalignment (e.g., **the shorter motion cannot synchronize with the original motion temporally**), which is a limitation we plan to address in future iterations.
>
> ***W2 & Q5: Benchmark Validation and Reliability.***
>
> We agree that validating the proposed metric is crucial. Following the protocol of MuDI [1], we conducted a correlation study between human evaluation and the metrics (C&C score and original Motion Fidelity ). We generated 120 videos across four methods (DreamBooth, VACE, CoMo w/o DAM, and CoMo) and asked human raters to evaluate "Compositional Motion Fidelity" (Success/Fail). We calculated Spearman’s rank correlation and AUROC between human scores and automatic metrics. As shown below, our C&C score aligns significantly better with human perception  than standard fidelity metrics, proving its reliability. Note that Spearman’s rank correlation score was computed using the normalized sum of all human evaluation answers(e.g., if 3 out of 5 raters answered "Success" the score is 0.6). The AUROC was computed based on the majority voting results (0 or 1) . Besides we provide qualitative examples **in  Figure 17.**
>
> |          Metric          | Spearman's Rank Correlation | AUROC |
> | :----------------------: | :-------------------------: | :---: |
> |     C&C Score (Ours)     |            0.50             | 0.71  |
> | Original Motion Fidelity |            0.19             | 0.56  |

---

> > ### Author Response · Authors · 2025-11-25
> >
> > ***W3 & Q1: Data Composition and Usage.***
> >
> > We apologize for the omitted details and have updated the paper:
> >
> > - **Training Data (Phase 1):** Phase 1 is a one-shot tuning process. We do *not* pretrain on large datasets. For each experiment, the training data is simply the single reference video provided by the user.
> > - **Benchmark Data:** We evaluate our framework on a curated benchmark of **23 high-quality reference videos** ( from previous studies like dynamic concepts[2] and  platforms like TikTok), characterized by high kinematic diversity. The dataset is primarily grounded in **human-centric videos** (19 clips),as these motion exhibits intricate articulation and complexity—ranging from diverse **street dancing** and **athletic sports** to challenging dynamic poses like **handstands**. The videos included two kinds of temporal length- 25 and 49 frames. These complex sequences rigorously test the model's ability to decouple dynamics from appearance. To further validate broad generalization beyond human subjects, our dataset also explicitly includes distinct  **2 object motions** (e.g., bear) and **2 camera movements** (e.g., circling views). For multi motion setting, we constructed **18  compositional pairs** (e.g., "Burpee + Dance").  Note the composed motions have the same temporal length for simplicity (49 frames).
> > - **Evaluation**: To compare our method in single setting, we generated 5 videos per motion (115 videos total) for testing.  In multi motion setting,  we   generated 5 videos per pair (90 videos total). The full dataset, along with the prompt list, will be open-sourced.
> >
> >
> >
> > ***Q3: Robustness to Viewpoint  and  temporal length***
> >
> > -  viewpoint: we  clarify that "camera motion in Section4.1" is in the single motion customization  setting, our method captures the entire temporal dynamic, whether object or camera movement. We explicitly evaluate camera motion ( in Fig. 6 (view translation) and Appendix Fig. 12 ("fast circling counterclockwise").Besides, we provide a novel 360-degree panoramic camera view customizatio**n in Fig13**.  While we can transfer these viewpoint changes, composing two *conflicting* global camera motions (e.g., pan left vs. pan right) is geometrically impossible in a single video, which is a natural physical constraint.
> > - temporal length:  composing motions of vastly different lengths may lead to temporal misalignment, as discussed in the W1 & Q2.
> >
> >
> >
> > ***Q4: Region Partitioning Strategy.***
> >
> > Currently, the partitioning is user-defined via a bounding box layout, similar to layout-to-image generation controls. This design was chosen to provide users with precise control over where actions occur (e.g., "I want the lion on the left and the bear on the right"). While an adaptive or learned partition (e.g., predicted by an LLM) is indeed a promising direction for fully automated generation, we argue that **highly variable or dynamic partitioning strategies often conflict with the intrinsic temporal dynamics of the motion itself**, potentially compromising motion integrity. Consequently, we adopted the current straightforward approach to explicitly confine motion within designated regions to ensure stability. We appreciate this insightful suggestion and plan to explore robust adaptive partitioning methods in future work; meanwhile, the current explicit control mechanism ensures high flexibility and reliability for the compositional task.
> >
> > ***Q6 Generalization across base models.***
> >
> > The *methodology* (Two-phase: Decoupled Tuning + Divide-and-Merge) is completely transferable to any DiT-based video model. However, the *trained LoRA weights* themselves are architecture-specific (dependent on the dimension of $W_q, W_k, W_v$ matrices). Transferring a specific trained motion module from Wan-2.1 to a different model (CogVideoX) would require rerunning Phase 1 on the new backbone.
> >
> > [1] Jang, Sangwon, et al. "Identity decoupling for multi-subject personalization of text-to-image models."   NeurIPS 2024
> >
> > [2] Abdal, Rameen, et al. "Dynamic concepts personalization from single videos."  SIGGRAPH 2025
> >
> > ------
> >
> > We hope these clarifications address your concerns, particularly regarding the metric validation and data details. We are happy to engage in further discussion.

---

### Author Response · Authors · 2025-11-25

Dear Reviewers,

We would like to sincerely thank you for  your constructive feedback and  recognizing the novelty of **CoMo** in addressing the motion-appearance entanglement and multi-motion blending challenges. We are encouraged that reviewers found our method achieves **state-of-the-art performance** (R1, R2) and our **structure clear** (R1, R2, R4).  In this revision, we have updated the following content:

1. **Metric Validation:**  We conducted a correlation study (Spearman’s rank correlation coefficient and AUROC)  verifying that our C&C metric aligns with human perception [**Lines 966-971 1003-1006**].

2. **More Baselines:** We added comparisons with MOFT and MotionClone[**Lines 1018-1025**] .

3. **Benchmark Details:** We provided full statistics on the dataset composition [**Lines 721-732**].

**Table: Quantitative  comparasion with more existing methods in single motion customization  (higher is better)**

|             | Text Sim  | Motion Fid | Temp. Cons. |
| :---------: | :-------: | :--------: | :---------: |
|  **Ours**   | **0.470** | **0.865**  |  **0.967**  |
| MotionClone |   0.436   |   0.782    |    0.917    |
|    MOFT     |   0.437   |   0.540    |    0.966    |

**Table: computational analysis in single motion setting**

**Table: computational analysis in single motion setting**

|                        | MotionDirector | DeT  | DreamBooth | MotionClone | MOFT | Ours |
| :--------------------: | :------------: | :--: | :--------: | ----------- | :--: | ---- |
| **Inference Time (s)** |      168       | 440  |    150     | 804         | 576  | 150  |
|   **Training or Guidance GPU Memory (G)**   |      31.1      | 30.9 |    26.5    | 51.5 (guidance)      |  75（guidance） | 31.5 |


**Table: computational analysis in compositional motion setting**

|                        | **1 motion** | **2 Motions** | **3 Motions** | **4 Motions** |
| ---------------------- | :----------: | :-----------: | :-----------: | :-----------: |
| **Inference Time (s)** |     150      |      194      |      276      |      360      |


**Table: Correlation between metrics and human evaluation.**

|                             | C&C score | original Motion Fidelity |
| :-------------------------: | :-------: | :----------------------: |
| Spearman's rank correlation |   0.50    |           0.19           |
|            AUROC            |   0.71    |           0.56           |

Regarding other concerns and questions you have raised, we have provided detailed and point-to-point responses under the corresponding comments.

We look forward to your feedback on our responses, and we also welcome you to raise new questions and engage in discussions.

Sincerely,

CoMo Authors

---

### Meta-Review · Area_Chair_FS3s · 2025-12-25

**Summary:**

Reviewers initially raised concerns regarding the lack of benchmark details, limited scalability in motion interaction, and questionable visual quality relative to the high optimization cost. During the rebuttal, while the authors provided dataset statistics and metric correlations that satisfied some reviewers, critical issues remain. Specifically, the trade-off between intensive per-video tuning and the resulting visual fidelity is seen as sub-optimal. Furthermore, errors in interpreting standard temporal consistency metrics and the inability to handle complex physical interactions suggest the work requires further refinement. I therefore recommend Reject.

**Reviewer Concerns:**

Addressed during Rebuttal:

- Authors clarified the composition of the 23-video benchmark and training protocols, satisfying the basic requirements for reproducibility.

- The distinction between the proposed DAM strategy and simple linear LoRA merging was clarified through additional visualizations.

- A human correlation study was provided to justify the new C&C metric.

Outstanding/Critical Points:

- The method requires thousands of training steps for a single video, yet the visual quality is not significantly superior to faster methods, raising doubts about the efficiency of the "overfitting" approach.

- Reviewers identified technical errors in the authors' analysis of VBench consistency metrics, which undermines the credibility of the quantitative evaluation.

- The framework is limited to spatially separated motions and fails to handle deep physical interactions or temporal misalignments between different motion lengths.

**Reviewer Scores:**

Reviewer p3ci (Rating: 4 $\to$ 4): While the benchmark details were helpful, the concerns regarding scalability and the inherent limitations of the spatial-split strategy remain largely unaddressed, likely keeping the score below the threshold.

Reviewer w24q (Rating: 6 $\to$ 6): This reviewer was the most positive and explicitly stated that their concerns were resolved. A slight upgrade or a firm maintain is expected.

Reviewer xkaZ (Rating: 4 $\to$ 4): This reviewer remained highly critical after the rebuttal, specifically pointing out errors in metric reporting and questioning the visual quality relative to the training effort. A score decrease or a firm reject recommendation is likely.

Reviewer kKY8 (Rating: 2 $\to$ 2 or 4): Although benchmark details were added, the reviewer’s core concern about the lack of comparison with sophisticated latent merging baselines was not fully resolved.

---

### Decision · Program_Chairs · 2026-01-26

Reject